**Investigation**

# Fdo1, Fkh1, Fkh2, and the Swi6–Mbp1 MBF complex regulate Mcd1 levels to impact eco1 rad61 cell growth in Saccharomyces cerevisiae

Gurvir Singh,[1] Robert V. Skibbens ID [1],*

[1]Department of Biological Sciences, Lehigh University, Bethlehem, PA 18015, USA

*Corresponding author: Department of Biological Sciences, Lehigh University, 111 Research Drive, Bethlehem, PA 18015, USA. Email: rvs3@lehigh.edu

Cohesins promote proper chromosome segregation, gene transcription, genomic architecture, DNA condensation, and DNA damage repair. Mutations in either cohesin subunits or regulatory genes can give rise to severe developmental abnormalities (such as Robert Syndrome and Cornelia de Lange Syndrome) and also are highly correlated with cancer. Despite this, little is known about cohesin regulation. Eco1 (ESCO2/EFO2 in humans) and Rad61 (WAPL in humans) represent two such regulators but perform opposing roles. Eco1 acetylation of cohesin during S phase, for instance, stabilizes cohesin-DNA binding to promote sister chromatid cohesion. On the other hand, Rad61 promotes the dissociation of cohesin from DNA. While Eco1 is essential, ECO1 and RAD61 co-deletion results in yeast cell viability, but only within a limited temperature range. Here, we report that eco1 rad61 cell lethality is due to reduced levels of the cohesin subunit Mcd1. Results from a suppressor screen further reveals that FDO1 deletion rescues the temperature-sensitive (ts) growth defects exhibited by eco1 rad61 double mutant cells by increasing Mcd1 levels. Regulation of MCD1 expression, however, appears more complex. Elevated expression of MBP1, which encodes a subunit of the MBF transcription complex, also rescues eco1 rad61 cell growth defects. Elevated expression of SWI6, however, which encodes the Mbp1-binding partner of MBF, exacerbates eco1 rad61 cell growth and also abrogates the Mpb1-dependent rescue. Finally, we identify two additional transcription factors, Fkh1 and Fkh2, that impact MCD1 expression. In combination, these findings provide new insights into the nuanced and multi-faceted transcriptional pathways that impact MCD1 expression.

Keywords: Roberts Syndrome (RBS); Cornelia de Lange Syndrome (CdLS); cohesins; ECO1/ESCO2; Rad61/WAPL; Mcd1/Scc1/RAD21; MBF (Mbp1 and Swi6); Fdo1; Fkh1; Fkh2

## Introduction

The biological functions of cohesins include DNA loop extrusion, sister chromatid tethering, chromosome compaction, and DNA damage repair (Guacci *et al.* 1997; Michaelis *et al.* 1997; Sjögren and Nasmyth 2001; Jessberger 2002; Kim *et al.* 2002; de Wit *et al.* 2015; Davidson *et al.* 2016). These chromatin conformations, however, vary extensively over the cell cycle. During G1 phase, for instance, cohesins produce both megabase sized epigenetically-defined topologically associating domains (TADs) and also smaller loops that alter the registration of regulatory elements (enhancer, promotors, insulator, etc.) (Rollins *et al.* 1999; Dorsett *et al.* 2005; Wendt *et al.* 2008; Degner *et al.* 2009, 2011; Chien *et al.* 2011; Seitan *et al.* 2011; de Wit *et al.* 2015; Davidson *et al.* 2016, 2019; Gassler *et al.* 2017; Haarhuis *et al.* 2017; Rao *et al.* 2017; Schwarzer *et al.* 2017; Wutz *et al.* 2017; Cuadrado *et al.* 2019; Kim *et al.* 2019; Zhang *et al.* 2019; Li *et al.* 2020; Xiang and Koshland 2021; Hsieh *et al.* 2022; Valton *et al.* 2022; Liang *et al.* 2023; Sept *et al.* 2023). Cohesin-based DNA loops, and loop resolution, are highly dynamic which could allow cells to efficiently respond to changes in external cues (nutrients, hormones, temperature, etc.) (Giorgetti *et al.* 2014; Hansen *et al.* 2017; Gabriele *et al.* 2022; Mach *et al.* 2022). During S phase, however, cohesins perform very different functions in that they tether together sister chromatids. This tethering is non-dynamic such that cohesins are stably bound until anaphase, when cohesion between sisters is inactivated to allow sister chromatids to segregate into the newly forming daughter cells (Guacci *et al.* 1997; Michaelis *et al.* 1997; Uhlmann and Nasmyth 1998; Skibbens *et al.* 1999).

The mechanisms that underlie cohesin activities remain hotly debated, although many of the fundamental aspects of cohesin complex assembly are known. For instance, budding yeast cohesin core components consist of Smc1, Smc3, and Mcd1 (Scc1). Smc1 and Smc3 each fold to create two extended intramolecular antiparallel coiled coils that bind to one another at their hinge-like domains (Haering *et al.* 2002; Gruber *et al.* 2003). Distal to the hinges, globular domains of Smc1 and Smc3 form ATPases (Haering *et al.* 2002; Arumugam *et al.* 2003). These ATPase head domains are capped by Mcd1 (Gligoris *et al.* 2014). The Mcd1 cap of this presumed core complex in turn recruits Irr1 (Scc3), Pds5, and Rad61 (Wpl1) (Hartman *et al.* 2000; Panizza *et al.* 2000; Losada *et al.* 2005; Rowland *et al.* 2009; Sutani *et al.* 2009; Gause *et al.* 2010; Kulemzina *et al.* 2012; Zhang *et al.* 2013; Gligoris *et al.* 2014; Roig *et al.* 2014; Tong and Skibbens 2014; Orgil *et al.* 2015; Muir *et al.* 2016). The mechanisms through which cohesins tether together sister chromatids (as one, two, or a cluster of cohesins

associated with each sister) or extrude DNA (so that both strands are simultaneously extruded into a growing loop) remain largely unknown (Haering *et al.* 2002; Zhang *et al.* 2008).Cohesin functions across the cell cycle are regulated both by auxiliary factors as well as by modifications of cohesin subunits. Relevant to the current study, Eco1/Ctf7 in budding yeast (Eso1 in fission yeast, Eco/Deco in *Drosophila*, CTF7 in *Arabidopsis*, and EFO1/ESCO1 and EFO2/ESCO2 paralogs in vertebrates) comprises a highly conserved family of essential acetyltransferases that regulate cohesins throughout the cell cycle (Skibbens *et al.* 1999; Toth *et al.* 1999; Tanaka *et al.* 2000; Williams *et al.* 2003; Vega *et al.* 2005; Seitan *et al.* 2006; Kawauchi *et al.* 2009; Mönnich *et al.* 2011; Whelan *et al.* 2012). During DNA replication, Eco1 is recruited to the replication fork by PCNA to acetylate Smc3 (Skibbens *et al.* 1999; Moldovan *et al.* 2006; Rowland *et al.* 2009; J. Zhang *et al.* 2017; W. Zhang *et al.* 2017; Bender *et al.* 2020). This modification converts the cohesin complex to a stable form. During G1, however, Eco1/ESCO family members help regulate DNA loop lengths (Alomer *et al.* 2017; Wutz *et al.* 2020; Van Ruiten *et al.* 2022). Rad61 (human WAPL), a cohesin-associated component, promotes the dissociation of unacetylated cohesin from DNA (Gandhi *et al.* 2006; Kueng *et al.* 2006; Rowland *et al.* 2009; Sutani *et al.* 2009). In this respect, Rad61 DNA-dissociating activity works in opposition to Eco1-based stabilization of cohesin binding to DNA such that yeast co-deleted for both remain viable (Ben-Shahar *et al.* 2008; Sutani *et al.* 2009; Maradeo and Skibbens 2010; Guacci and Koshland 2012). Cohesins also are regulated through one of the core subunits—Mcd1. Mcd1 is unique among cohesin subunits in that it is largely degraded at anaphase onset and must be newly transcribed each and every cell cycle (Guacci *et al.* 1997; Uhlmann *et al.* 1999).

Mutation of cohesin pathway genes can give rise to Robert Syndrome (RBS) and Cornelia de Lange Syndrome (CdLS). Individuals with RBS and CdLS manifest a multitude of overlapping developmental abnormalities that, when severe, result in premature mortality and terminal miscarriages (Krantz *et al.* 2004; Tonkin *et al.* 2004; Schüle *et al.* 2005; Vega *et al.* 2005; Musio *et al.* 2006; Deardorff *et al.* 2007; Gordillo *et al.* 2008). Defects in cohesin regulation are also tightly correlated with several forms of cancer (Antony *et al.* 2021; Di Nardo *et al.* 2022; Pati 2024). Despite the importance of cohesin functions, little is known regarding the regulation of cohesins early during the cell cycle (pre-S phase). Recent findings, however, suggest that *MCD1* expression may be a critical component of cohesin regulation throughout the cell cycle (Buskirk and Skibbens 2022; Choudhary *et al.* 2022). In this study, we identify *FDO1* as a novel regulator of *MCD1* expression. We further report that two additional transcriptional factors in the Forkhead box (Fkh1 and Fkh2 in yeast) family, as well as the Swi6–Mbp1 MBF transcription complex, play key roles in *MCD1* regulation and therefore likely impact all cohesin activities.

## Materials and methods
### Yeast strains, media, and growth conditions
All strains (Supplementary Table 1) were grown on YPD-rich media unless placed on selective medium to facilitate plasmid transformations or spore identification (Rose *et al.* 1990).

Phenotypic analysis was assessed as previously described with minor modifications (Buskirk and Skibbens 2022). Briefly, log phase cultures grown at the permissive temperature (30°C unless otherwise indicated) were normalized based on $OD_{600}$, serially 10-fold dilutions plated in duplicates on either YPD agar plates or selective medium plates, and then maintained across a range of temperatures (typically comparing growth at 30°C vs 37°C).

### Strain generation
The primers utilized to delete or partially delete genes, and then verify deletions, are listed in Supplementary Table 2. DNA products used to replace *FDO1*, *FKH1* and *FKH2* ORFs were obtained using PCR and a kanMX6 template as previously described (Longtine *et al.* 1998).

### Plasmid generation
Overexpression of selected genes was generated following a strategy previously described. In brief, DNA oligos (Supplementary Table 2) were designed to produce complete *FKH1*, *FKH2*, *FDO1*, *SWI6*, and *MBP1* ORFs (including roughly 300 base pairs upstream of the starting codons). PCR products and pRS424 (2μ TRP) or pRS425 (2μ LEU) plasmids were digested and ligated to generate the following plasmids: pGS7, pGS8 (*FKH1* and pRS424 digested with SacII-XhoI), pGS6 (*FKH2* and pRS424 digested with SacII/Xho1fragment), pGS11, pGS12, pGS13 (*FDO1* and pRS 424 digested with SacII/Xho1), pGS14, pGS15, pGS16 (*SWI6* and pRS 424 digested with SacII/XmaI), pGS31, pGS32, pGS33 (*MBP1* and pRS 425 digested with XhoI/SacII). The *MCD1* (2μ TRP and 2μ LEU) plasmids were generated by digesting (with ClaI for the 2μ TRP and XhoI—SacII for the 2μ LEU plasmid) the *MCD1* coding sequence along with its endogenous promotor from pVG185 (2μ URA, kindly provided by Dr. Vincent Guacci) and ligating it in pRS424. All generated plasmids were verified by restriction digest. The specific isolates of the plasmid transformed in the parent or wildtype cells are contained in the Supplementary Table 1 and the primers used to create the plasmids are enclosed in Supplementary Table 2.

### Genomic sequencing
Genemomic sequencing of the revertant strains was performed as previously described (Buskirk and Skibbens 2022).

### Western blot protein extraction and quantification
Cell numbers for each strain within an experiment were normalized by $OD_{600}$. Cells were then washed prior to suspending in 1.0 Sorbitol and exposed to Zymolyase (100T) (USBiological Life Sciences) and β-mercaptoethanol (1/50th of total volume) for 30 min at 37°C. Cell wall digestion was confirmed by assessing cell lysis microscopically, upon exposure to 0.5% SDS. Cell extracts were resolved by SDS PAGE, then transferred to a PVDF membrane. Post blocking (5% NFDM, 0.1% BSA, 1X PBS), proteins were detected using primary antibodies PGK (mouse) at 1:1,000 K, Invitrogen catalog number 459250; Mcd1 (Rabbit) at 1:20 K—kindly provided by Dr Vincent Guacci- followed by secondary antibodies Goat anti-Rabbit HRP at 1:10 K, BIO-RAD catalog number 170-6515; goat anti-Mouse HRP at 1:10 K, BIO-RAD catalog number 170-6516 and ECL Prime (Amersham, RPN2232) and Xray film (Denville Scientific, HyBlot ES™, catalog number E3218) development (Xomat) using the Konica Minolta SRX-101A film processor. Protein band intensities, obtained by film scanning (EPSONPERFECTION V300 PHOTO) were quantified using Image J. Significance was determined by a two-tailed test (P-value less than 0.05).

### Flow cytometry and cell cycle progression
Prior to generating cell extracts, all strains within an experiment were maintained in log growth over a two day regimen as previously described (Buskirk and Skibbens 2022). Briefly, log phase

cultures were then normalized (OD$_{600}$, typically to achieve 1.0 OD of cells) and then synchronized in early S phase by the addition of hydroxyurea (SIGMA, H8627) at final concentration of 0.2 M. Log phase growth and cell cycle arrest in S phase was confirmed by flow cytometry (BD FACScan) as previously described (Maradeo and Skibbens 2010; Tong and Skibbens 2015).

## Results

### Reduced Mcd1 levels are responsible for *eco1Δ rad61Δ* double mutant yeast cell temperature-sensitive inviability

Eco1 (which is essential only during S phase) and Pds5 (an auxiliary cohesin subunit required for cohesion from S phase to anaphase onset) support cohesin functions through independent mechanisms (Skibbens *et al.* 1999; Toth *et al.* 1999; Hartman *et al.* 2000; Panizza *et al.* 2000). In a remarkable convergence of studies, however, deletion of *CLN2* was found to promote the viability of both *eco1Δ rad61Δ* at 37°C and *pds5Δ elg1Δ* cells (Buskirk and Skibbens 2022; Choudhary *et al.* 2022). The latter study provided evidence that *CLN2* deletion increased Mcd1 levels, such that simply elevating Mcd1 protein levels were sufficient to render viable *pds5Δ elg1Δ* double mutant cells (Choudhary *et al.* 2022). Given that *CLN2* is common to both *eco1Δ rad61Δ* and *pds5Δ elg1Δ* cells as a bypass suppressor (Buskirk and Skibbens 2022; Choudhary *et al.* 2022), it became important to test whether Mcd1 protein levels are similarly reduced in *eco1Δ rad61Δ* cells, even at permissive temperatures. To test this possibility, Mcd1 levels from wildtype and *eco1Δ rad61Δ* cells were quantified by Western Blots. Given that Mcd1 levels rise during S phase and fall precipitously during anaphase (Guacci *et al.* 1997), log phase cultures were arrested in early S phase (hydroxy urea, HU) and samples harvested to assess DNA content (flow cytometry) and Mcd1 levels (Western blot using Mcd1-directed antibody—a generous gift from Dr. Vincent Guacci) (Guacci *et al.* 1997). As expected, all log phase cultures contained both 1C and 2C DNA peaks that collapsed to an early stage of DNA replication in response to HU (Fig. 1a). To facilitate quantification by Western blot, we first performed a dilutions series to identify a linear range of detection for both Mcd1 and Pgk1 (loading control) across each of our three biological replicates (Fig. 1b,c). Appropriately diluted extracts were then assessed. As expected, wildtype cells arrested in the S phase contained high levels of Mcd1 (Fig. 1d), relative to other portions of the cell cycle (data not shown). In contrast, Mcd1 levels were significantly decreased ($P = 0.0031$) in S phase-arrested *eco1Δ rad61Δ* cells, compared to wildtype cells (Fig. 1e). Mcd1 protein levels fell to nearly undetectable levels upon shifting to 37°C (data not shown), at which point *eco1Δ rad61Δ* cells are inviable.

If the reduction of Mcd1 protein is the sole basis for *eco1Δ rad61Δ* cell inviability during thermal stress, then exogenously elevating Mcd1 levels should rescue *eco1Δ rad61Δ* cell temperature-sensitive (ts) growth. To test whether the increase in Mcd1 protein alone is sufficient to rescue *eco1Δ rad61Δ* cell temperature sensitivity, we generated an *MCD1* high-copy (2μ *TRP1*) plasmid. We validated this reagent by confirming that it indeed rescued the ts-growth of *mcd1-1* mutated cells (Fig. 2a). Log phase wildtype and *eco1Δ rad61Δ* cells, transformed with vector alone or vector overexpressing *MCD1*, were serially diluted, plated onto selective media, and then incubated at either 30°C or 37°C. Cells are apparently unaffected by elevated Mcd1 levels in that both wildtype and *eco1Δ rad61Δ* cells exhibited robust growth at 30°C (Fig. 2b). As expected, *eco1Δ rad61Δ* cells that contained vector alone exhibited severe growth defects at 37°C.

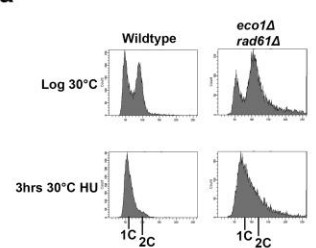

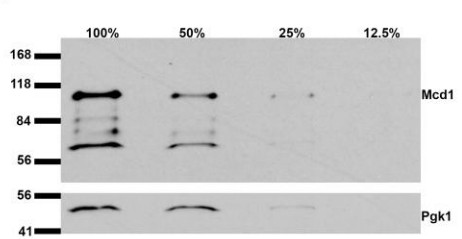

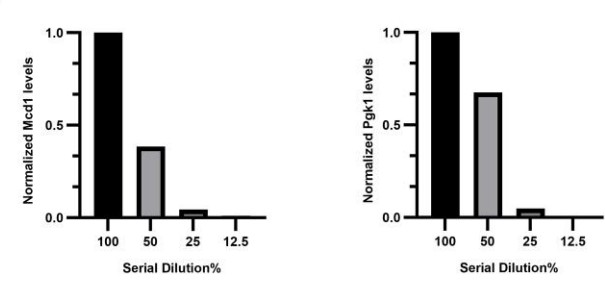

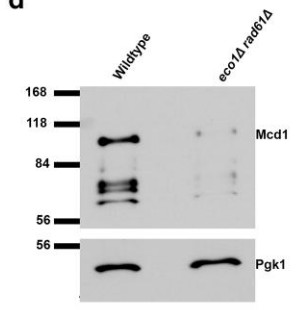

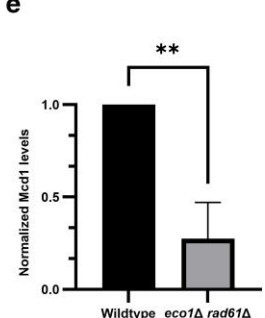

**Fig. 1.** *eco1Δ rad61Δ* double mutant cells contain reduced Mcd1 levels. a) Flow cytometry data of DNA contents for wildtype (YBS255) and *eco1Δ rad61Δ* double mutant cells (YMM828). b) Representative Western Blot of Mcd1 (top panel) and Pgk1 (lower panel) protein levels obtained from serially diluted (100%, 50%, 25%, 12.5%) extracts of HU-synchronized wildtype cells indicated in (A). c) Quantification of Mcd1 (left panel) and Pgk1 (right panel) of the serially diluted (100%, 50%, 25%, 12.5%) sample in (B). d) Representative Western Blot of Mcd1 (top panel) and Pgk1 (lower panel) protein levels of 50% diluted wildtype and *eco1 rad61* null cell extracts obtained from HU-synchronized cells indicated in (A). e) Quantification of Mcd1, normalized to Pgk1 loading controls. Statistical analysis was performed using a two-tailed *t*-test. Statistical differences (**) are based on a $P < 0.01$ obtained across three experiments ($n = 3$). $P = 0.0031$ for *eco1Δ rad61Δ* compared to wildtype cells. Error bars indicate the standard deviation.

*eco1Δ rad61Δ* cells that harbored the *MCD1* overexpression vector, however, exhibited robust growth at 37°C (Fig. 2b). These findings reveal that *eco1Δ rad61Δ* cells contain reduced Mcd1 protein, even at permissive temperatures, and that elevating Mcd1

protein levels is solely sufficient to rescue *eco1Δ rad61Δ* cell lethality at otherwise non-permissive temperatures.

## Identification of Fdo1 as a novel regulator of cohesin

Previously, a genome-wide suppressor screen identified the G1 cyclin Cln2 as a novel regulator of *eco1Δ rad61Δ* mutant cells (Buskirk and Skibbens 2022). To further identify novel genes that rescue *eco1Δ rad61Δ* temperature sensitivity, a second revertant (YBS1638) was back-crossed 2 times to wildtype cells and multiple non-ts *eco1Δ rad61Δ* segregants pooled together prior to whole genome sequencing as previously described (Buskirk and Skibbens 2022). Mutations that were common across all segregants (present at a frequency of 1.0) were prioritized as revertant gene candidates that rescued the temperature sensitivity otherwise exhibited by *eco1Δ rad61Δ* cells. In contrast, non-revertant mutations were expected to occur at frequencies below 1.0. Only one gene, *FDO1*, was mutated (encoding only the first 82 amino acids, herein fdo1$^{1–82Δ}$) in all segregants, although mutation of *RPL31A* was

also present at a high frequency (Table 1). Other gene mutations (*MSH5*, *ARN1*, and *EAF1*) were present at significantly reduced frequencies, eliminating them as likely reverting candidate genes. We decided to delete *FDO1* de novo from parent *eco1Δ rad61Δ* cells to test for the rescue of ts-growth defects in the absence of other mutations. We generated multiple isolates of *eco1Δ rad61Δ* cells either deleted for the complete *FDO1* coding sequence, or the partial deletion that arose in the revertant spontaneous suppressor screen (fdo1$^{1–82Δ}$). Log phase cultures of the wildtype, parental *eco1Δ rad61Δ* cells, and three independent *eco1Δ rad61Δ fdo1Δ* triple null cells were serially diluted, plated onto YPD agar, and incubated at either 30°C or 37°C. As expected, all strains exhibited robust growth at 30°C while *eco1Δ rad61Δ* cells were inviable at 37°C (Fig. 3a). In contrast, all three isolates of *eco1Δ rad61Δ fdo1Δ* triple null cells were viable at 37°C (Fig. 3a). A similar rescue of ts-growth was obtained for isolates (two shown) of *eco1Δ rad61Δ* cells that harbored truncated Fdo1 protein (fdo1$^{1–82Δ}$) (Fig. 3b). These results confirm that *FDO1* loss-of-function rescues the ts-growth defects otherwise present in *eco1Δ rad61Δ* double mutant cells.

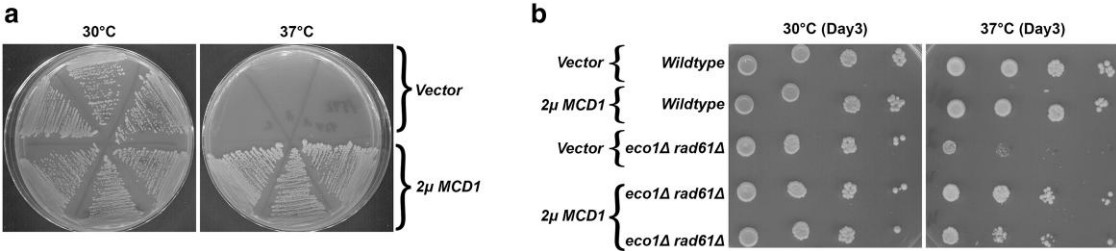

**Fig. 2.** *MCD1* overexpression rescues *eco1 rad61* double mutant cell ts-growth. a) Streak test of three independent isolates of an *mcd1-1* temperature-sensitive strain (YMM396) transformed with vector alone, compared to three independent isolates of *mcd1-1* ts cells transformed with vector overexpressing *MCD1*. Temperature growth conditions are indicated. b) Growth of 10-fold serial dilutions of wildtype cells overexpressing vector alone (YGS26) or overexpressing *MCD1* (YGS27), and *eco1Δ rad61Δ* double mutant cells overexpressing vector alone (YGS28), or overexpressing *MCD1* (2 isolates shown; YGS29, YGS30). Temperatures and days of growth are indicated.

**Table 1.** Identification of spontaneous revertant mutations.

| Chromosome | Position | DNA change | Protein change | Coding impact | Gene | Description from SGD$^a$ | Frequency |
|---|---|---|---|---|---|---|---|
| chrXIII | 553608 | 247C>T | Nonsense at Q83 | nonsense | FDO1 (YMR144W) | Protein involved in directionality of mating-type switching | 1.00 |
| chrIV | 322263 | 38C>T | T13I | missense | RPL31A (YDL075W) | Ribosomal 60S subunit protein L31A | 0.91 |
| chrIV | 178744 | 411G>T | W137C | missense | MSH5 (YDL154W) | Protein of the MutS family | 0.59 |
| chrVIII | 19897 | 1075G>C | A359P | missense | ARN1 (YHL040C) | ARN family transporter for siderophore-iron chelates | 0.73 |
| chrIV | 1193539 | 1346T>C | I449T | missense | EAF1 (YDR359C) | Component of the NuA4 histone acetyltransferase complex | 0.24 |

$^a$Description of gene from SGD.

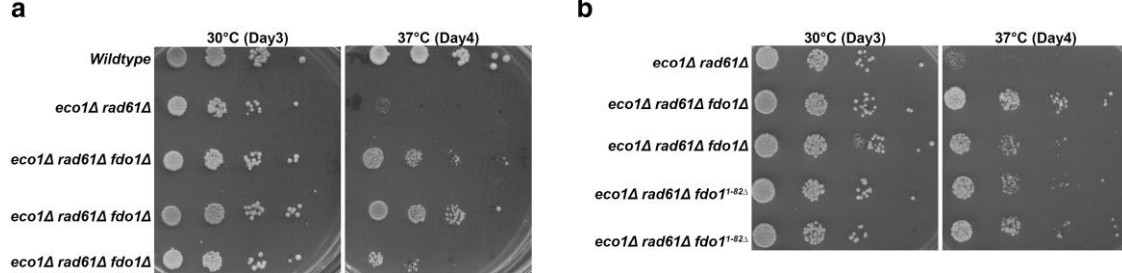

**Fig. 3.** Identification of *FDO1* deletion as a suppressor of *eco1Δ rad61Δ* cell temperature-sensitive growth defects. a) Growth of 10-fold serial dilutions of wildtype (YBS255), *eco1Δ rad61Δ* double mutant cells (YMM828) and three independent isolates of *eco1Δ rad61Δ fdo1Δ* triple mutant cells (YGS4, YGS36, YGS37). Temperatures and days of growth are indicated. b) Growth of 10-fold serial dilutions of *eco1Δ rad61Δ* double mutant cells (YMM828), two independent isolates of *eco1Δ rad61Δ fdo1Δ* triple mutant cells (YGS3, YGS1), and two independent isolates of *eco1Δ rad61Δ fdo1$^{1–82Δ}$* (YGS5, YGS6) triple mutant cells. Temperature and days of growth are indicated.

## Fdo1 regulates *MCD1* expression in *eco1Δ rad61Δ* double mutant

Deletion of *FDO1* rescues *eco1Δ rad61Δ* cell ts-growth defects (Fig. 2). It thus became important to test whether deletion of *FDO1* increased Mcd1 protein levels in *eco1Δ rad61Δ* cells. Whole cell extracts from *eco1Δ rad61Δ fdo1Δ* triple mutant cells, obtained from each of the three biological replicates arrested in early S phase (Fig. 4a), were diluted and probed by Western blot to obtain Mcd1 and Pgk1 signals that fell within a linear range (Fig. 4b, c). Appropriately diluted extracts from *eco1Δ rad61Δ fdo1Δ* cells were then compared to extracts obtained from *eco1Δ rad61Δ* cells (Fig. 4d). The results reveal that Mcd1 levels are increased significantly (P = 0.0257) across the *eco1Δ rad61Δ fdo1Δ* triple mutant isolates, compared to *eco1Δ rad61Δ* cells (Fig. 4d, e). In combination, these findings document that *eco1Δ rad61Δ* cells contain low levels of Mcd1 and that *FDO1* deletion rescues the temperature sensitivity of the *eco1Δ rad61Δ* double mutant cells at least in part by increasing Mcd1 levels.

If *FDO1* deletion rescues *eco1Δ rad61Δ* cell ts-growth defects by increasing Mcd1 protein levels, we reasoned that elevated Fdo1 levels might produce adverse growth defects. To test this possibility, *FDO1* was inserted into a high-copy plasmid (2μ *TRP1*) and transformed into wildtype and *eco1Δ rad61Δ* cells. Log phase cultures of the resulting strains were serially diluted onto selective media plates and incubated at either 30°C or 37°C. While elevated *FDO1* had no effect on wildtype cells, *eco1Δ rad61Δ* cells that harbor exogenous *FDO1* constructs were largely inviable even at 30°C (Fig. 5a). While these results provide strong evidence that Fdo1 acts as a repressor of *MCD1* expression, we realized that elevated Fdo1 levels likely alters the transcription of genes beyond *MCD1*, which could account for the observed cell lethality. If the lethal effect produced by elevated Fdo1 is independent of *MCD1* expression in *eco1Δ rad61Δ* cells, then co-expressing high-copy constructs of both *FDO1* and *MCD1* should fail to rescue that cell lethality. To test this possibility, we transformed *eco1Δ rad61Δ* cells with a combination of vectors alone and vectors driving elevated expression of either *FDO1*, *MCD1*, or both *FDO1 and MCD1*. Log phase cultures of the resulting strains were serially diluted onto selective media plates and incubated at either 30°C or 37°C. As expected, elevated expression of *MCD1* alone rescued *eco1Δ rad61Δ* cell ts-growth while elevated expression of *FDO1* greatly exacerbated the ts-growth defect (Fig. 5b). Notably, *eco1Δ rad61Δ* cells that harbored both *MCD1* and *FDO1* high-copy plasmids exhibited a partial rescue of the cell ts-growth defects (Fig. 5b). This somewhat attenuated rescue likely reflects a level of inhibition exerted by elevated Fdo1 levels on the native *MCD1* promotor present on the high-copy plasmid. In combination, these and the preceding results reveal that Fdo1 acts to repress *MCD1* expression and to a level that is lethal in *eco1Δ rad61Δ* cells during thermic stress.

## Forkhead proteins, Fkh1 and Fkh2, play additional roles in *MCD1* regulation

*FDO1* was originally identified as prohibiting misdirected mating-type switching (Dummer *et al.* 2016). Biochemical studies further revealed that Fdo1 interacts with the FOX-type transcription factor Forkhead 1 (Fkh1) (Ho *et al.* 2002; Dummer *et al.* 2016), which similarly inhibits misdirected mating-type switching and also negatively regulates transcriptional control of cell cycle progression (Dummer *et al.* 2016; Aref *et al.* 2021). It thus became

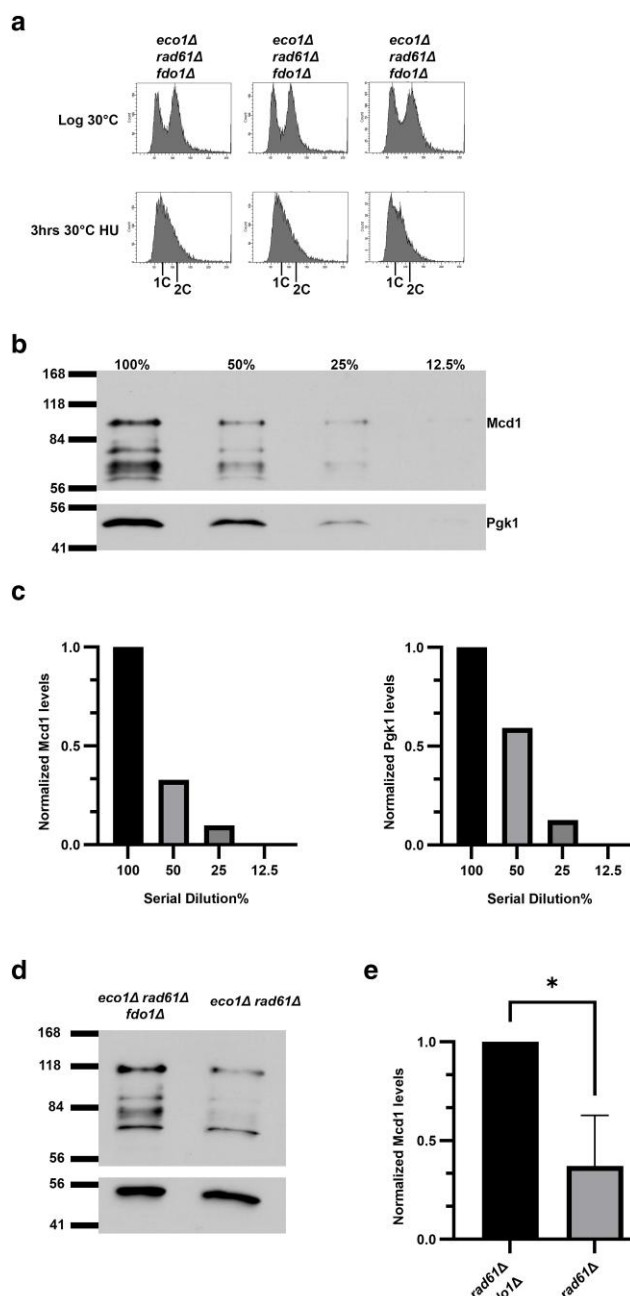

**Fig. 4.** Deletion of *FDO1* elevates Mcd1 levels in *eco1Δ rad61Δ* double mutant cells. a) Flow cytometry data of DNA content for three independent isolates of *eco1Δ rad61Δ fdo1Δ* triple mutant cells (YGS3, YGS4, YGS31). Log phase and early S phase synchronized (HU, hydroxyurea) DNA profiles are shown for each strain. b) Representative Western Blot of Mcd1 (top panel) and Pgk1 (lower panel) protein levels of the serially diluted (100%, 50%, 25%, 12.5%) *eco1Δ rad61Δ fdo1Δ* triple mutant cell extracts obtained from HU-synchronized cells indicated in (a). c) Quantifications of Mcd1 (left panel) and Pgk1 (right panel) of the serially diluted (100%, 50%, 25%, 12.5%) sample in (b). d) Representative Western Blot of Mcd1 (top panel) and Pgk1 (lower panel) protein levels of a 50% diluted *eco1Δ rad61Δ fdo1Δ* triple mutant and *eco1Δ rad61Δ fdo1Δ* double mutant cell extracts obtained from HU-synchronized cells indicated in (a). e) Quantification of Mcd1, normalized to Pgk1 loading controls. Statistical analysis was performed using a two-tailed *t*-test. Statistical differences (*) are based on a P < 0.05 obtained across three experiments (n = 3). P = 0.0133 for *eco1Δ rad61Δ fdo1Δ* compared to *eco1Δ rad61Δ* cells. Error bars indicate the standard deviation.

important to test whether deletion of *FKH1* would rescue the temperature sensitivity of *eco1Δ rad61Δ* cells, similar to the deletion of *FDO1*. The entire *FKH1* ORF was deleted from *eco1Δ rad61Δ* cells and confirmed following established protocols (Longtine *et al.* 1998). Parental *eco1Δ rad61Δ* cells, and two independent *eco1Δ rad61Δ fkh1Δ* triple null cells were serially diluted, plated onto YPD agar, and incubated at either 30°C or 37°C. Surprisingly, the deletion of *FKH1* failed to revert the temperature sensitivity otherwise exhibited by *eco1Δ rad61Δ* cells (Fig. 6a). The budding yeast genome contains a paralog of Fkh1, termed Fkh2, that exhibits redundant transcriptional roles in cell cycle control (Zhou and Shi 2022). Thus, we considered the possibility that Fdo1 impacts cohesin function through Fkh2 instead of Fkh1. Similar to the above strategy, the entire *FKH2* ORF was deleted from *eco1Δ rad61Δ* cells. Serial dilution analyses, however, revealed that *FKH2* deletion provides no growth benefit to *eco1Δ rad61Δ* cells at 37°C. In fact, all independent isolates of *eco1Δ rad61Δ fkh2Δ* triple mutant cells

exhibited significantly exacerbated ts-growth defects at 37°C (Fig. 6b). In combination, these findings differentiate the role of Fdo1 from that of Fkh paralogs and further reveals a novel negative genetic interaction that further differentiates Fkh2 from Fkh1.

The negative effect observed by the deletion of *FKH2*, but not the close paralog encoded by *FKH1*, was surprising. To further investigate this link, we wondered whether *FKH2* deletion might exacerbate *eco1Δ rad61Δ* cell viability specifically by reducing Mcd1 levels. If correct, then the increased ts-growth defects should be rescued by exogenous *MCD1* overexpression. To test this prediction, *eco1Δ rad61Δ fkh2Δ* cells were transformed with vector alone or vector driving elevated expression of *MCD1*. Log phase cultures of the resulting transformants were serially diluted and plated onto selective media. Elevated expression of *MCD1* had no effect on *eco1Δ rad61Δ fkh2Δ* cells held at the permissive temperature of 30°C (Fig. 7). As expected, *eco1Δ rad61Δ fkh2Δ* cells transformed with vector alone exhibited severe growth defects at the

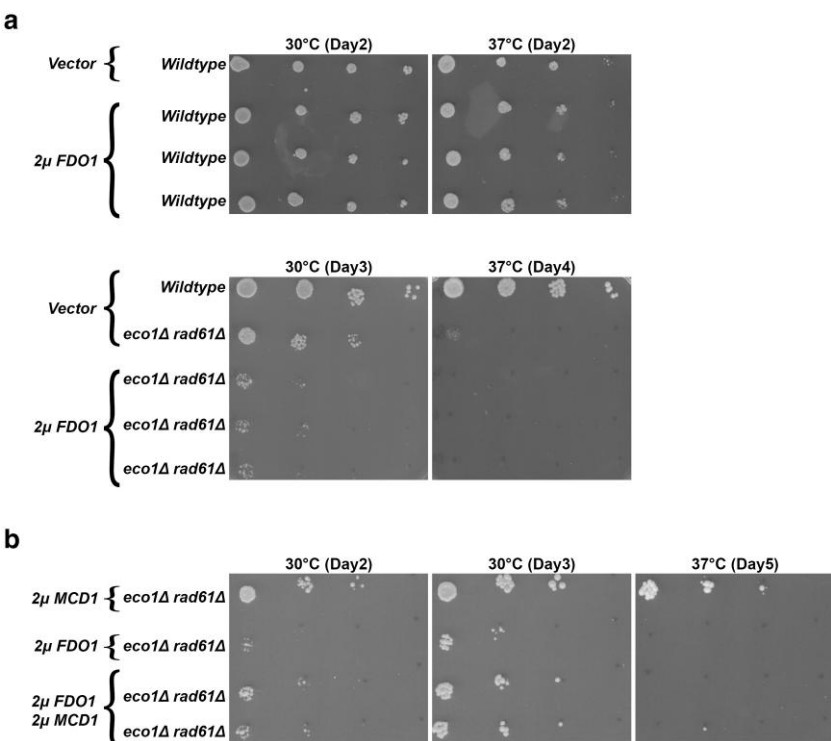

**Fig. 5.** Increased Mcd1 levels partially rescue the growth defect caused by elevated Fdo1 levels. a) Top panel: Growth of 10-fold serial dilutions of wildtype cells overexpressing vector alone (YGS26) and three independent isolates of wildtype cells overexpressing *FDO1* (YGS98, YGS100, YGS102). Temperatures and days of growth are indicated. Bottom panel: Growth of 10-fold serial dilutions of wildtype cells overexpressing vector alone (YGS26), *eco1Δ rad61Δ* double mutant cells overexpressing vector alone (YGS28), and three independent isolates of *eco1Δ rad61Δ* double mutant cells overexpressing *FDO1* (YGS99, YGS101, and YGS103). Temperatures and days of growth are indicated. b) Growth of 10-fold serial dilutions of *eco1Δ rad61Δ* double mutant cells (YMM828) overexpressing *MCD1* (YGS179), overexpressing *FDO1* (YGS193), and two independent isolates of *eco1Δ rad61Δ* co-overexpressing both *FDO1* and *MCD1* (YGS195, YGS196). Temperature and days of growth are indicated.

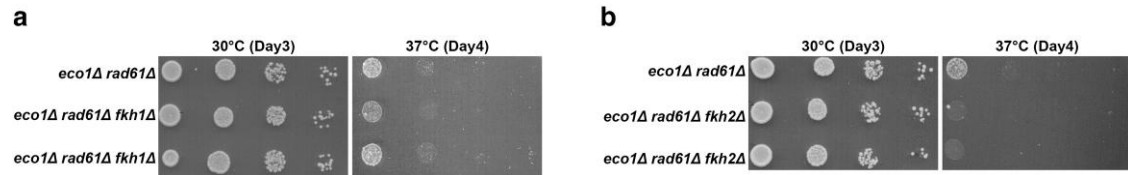

**Fig. 6.** *FKH2* deletion exacerbates *eco1Δ rad61Δ* cell ts-growth. a) Growth of 10-fold serial dilutions of *eco1Δ rad61Δ* double mutant cells (YMM828) and two independent isolates of *eco1Δ rad61Δ fkh1Δ* triple mutant cells (YGS 11, YGS12). Temperatures and days of growth are indicated. b) Growth of 10-fold serial dilutions of *eco1Δ rad61Δ* double mutant cells (YMM828) and two independent isolates of *eco1Δ rad61Δ fkh2Δ* triple mutant cells (YGS13, YGS14). Temperatures and days of growth are indicated.

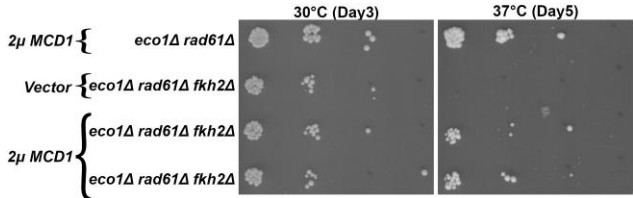

**Fig. 7.** *MCD1* overexpression rescues the deleterious effect of *FKH2* deletion from *eco1 rad61* null cells. Growth of 10-fold serial dilutions of *eco1Δ rad61Δ* doble mutant cells (YMM828) overexpressing *MCD1* (YGS29), *eco1Δ rad61Δ fkh2Δ* triple mutant cells overexpressing vector alone (YGS176), and two independent isolates of *eco1Δ rad61Δ fkh2Δ* overexpressing *MCD1* (YGS177, YGS178). Temperature and days of growth are indicated.

restrictive temperature of 37°C. In contrast, *MCD1* overexpression partially rescued the ts-growth defect otherwise exhibited by *eco1Δ rad61Δ fkh2Δ* cells (Fig. 7). In parallel, we transformed *eco1Δ rad61Δ fkh1Δ* cells with vector alone and vector driving elevated expression of *MCD1*. Here, we expected a full rescue since *FKH1* deletion does not overtly impact *eco1Δ rad61Δ* growth at 37°C. Indeed, *MCD1* overexpression fully rescued the ts-growth defects otherwise exhibited by *eco1Δ rad61Δ* (Supplementary Fig. 1). In combination, these findings reveal that Fkh2 plays a key role, and unique from that of Fkh1, in *MCD1* regulation during thermal stress.

Despite the adverse effect of deleting *FKH2*, we considered the possibility that co-deletion of both *FKH1* and *FKH2* might be required to rescue the ts-growth defects exhibited by *eco1Δ rad61Δ* cells. Diploid cells obtained by direct mating of *eco1Δ rad61Δ fkh1Δ* (YGS11) to *eco1Δ rad61Δ fkh2Δ* (YGS14), however, produced few tetrads upon sporulation. To circumvent this effect, we generated a diploid (YBS4450) homozygous for *ECO1* and *RAD61* deletions, heterozygous for *FKH1* and *FKH2* deletions, but that also harbored *ECO1* (linked to *LEU2* and integrated into chromosome III, independent of the native *ECO1* locus on chromosome VI). Since both *FKH1* and *FKH2* are replaced by the *kanMX6* selectable marker (Longtine *et al.* 1998), the final dispositions of *FKH* deletions were performed using PCR. Subsequent sporulations/dissections of 43 tetrads produced 19 *eco1Δ rad61Δ KAN⁺* spores, 6 of which were excluded from further analyses since *fkh1Δ* and *fkh2Δ* clearly segregated independent of one another across the four-spore tetrads. Of the remaining 13 spores, PCR confirmed that none of the *eco1Δ rad61Δ* cells contained both *fkh1Δ* and *fkh2Δ* (Supplementary Fig. 2). In contrast, we recovered roughly equal numbers of *eco1Δ rad61Δ fkh1Δ* (7) and *eco1Δ rad61Δ fkh2Δ* (6) spores. These results provide strong evidence that the combined deletion of both *FHK1* and *FKH2* not only fails to rescue ts-growth defects, but instead may be lethal in *eco1Δ rad61Δ* cells.

To further test the conclusion that co-deletions of *FKH* genes may be lethal in *eco1Δ rad61Δ* cells, we turned to *eco1Δ rad61Δ KAN⁺ ECO1:LEU2* isolates obtained from the above dissections. Did retention of *ECO1* allow for the recovery of *eco1Δ rad61Δ fkh1Δ fkh2Δ* mutant cells? As before, we excluded from further analyses tetrads in which *fkh1Δ* and *fkh2Δ* segregated independent of one another across four-spore tetrads. We then performed PCR on 20 of the remaining 53 spores. The results reveal that 8 of the 20 *eco1Δ rad61Δ* spores harbor deletions of both *FKH1* and *FKH2* (Supplementary Fig. 3), obviating concerns that the above analyses failed to identify quadruple deletions for technical reasons. Importantly, all eight spores were LEU+, providing additional support for the model that the *fkh1Δ fkh2Δ* combination is indeed lethal in *eco1Δ rad61Δ* cells.

The results provided throughout this study predict that the synthetic lethality observed in *eco1Δ rad61Δ fkh1Δ fkh2Δ* cells might be due to decreased levels of Mcd1. If correct, then elevated levels of *MCD1* should allow for the recovery of *eco1Δ rad61Δ fkh1Δ fkh2Δ*. To test this, the above diploid (YBS4450 harboring *eco1Δ rad61Δ fkh1Δ fkh2Δ ECO1:LEU2*) was transformed with the high-copy vector (*2μ TRP1*) that contains *MCD1*. The resulting strain (YBS4535) was sporulated and the ability to recover *eco1Δ rad61Δ fkh1Δ fkh2Δ 2μm-TRP1-MCD1 (without ECO1:LEU2)* assessed. Surprisingly, we recovered few TRP1⁺ spores (10 of 42 viable spores), two of which were devoid of *ECO1* (LEU⁻, HIS⁺) but retained kanMX6⁺. The first of these (YBS4547) exhibited normal growth at 30°C while the second (YB4548) required several additional days to form even a small colony (not shown). PCR analyses revealed that the first *eco1Δ rad61Δ 2μm-TRP1-MCD1* spore (YBS4547) retained only fhk2Δ. The second *eco1Δ rad61Δ 2μm-TRP1-MCD1* spore (YBS4548), however, retained both *fkh1Δ* and *fhk2Δ* (Supplementary Fig. 4). These results support the model that deletion of both *FKH1* and *FKH2* are lethal in *eco1Δ rad61Δ* due, in part, to reduced *MCD1* expression.

## Forkhead factors 1 and 2 both perform independent roles in cohesin regulation

Given that *fkh2Δ* alone produces adverse effects on *eco1Δ rad61Δ* cell growth, but that *fkh1Δ fkh2Δ* is lethal in *eco1Δ rad61Δ* cells, we considered the possibility that Fkh2 and Fkh1 might both promote *MCD1* expression. To test this, we generated high-copy (*2μ TRP1*) plasmids that contain either *FKH2* or *FKH1*. Log phase wild-type and *eco1Δ rad61Δ* cells, each transformed with vector alone or vector containing either *FKH2* or *FKH1*, were serially diluted and plated onto selective media. While elevated expression of *FKH2* had no effect on wildtype cells, elevated *FKH2* expression rendered *eco1Δ rad61Δ* cells largely inviable even at 30°C (Fig. 8a). *FKH1* elevated expression also produced *eco1Δ rad61Δ* cell growth defects at 30°C, but less severe than cells with elevated *FKH2* expression (Fig. 8b). In combination, these findings suggest that elevated levels of Fkh1 and Fkh2 each negatively impact cohesin pathways in an undetermined fashion.

Of the many scenarios that could explain the negative effect produced by elevated FHK proteins, we were intrigued by the possibility that reduced Mcd1 levels were involved. If this model is correct, then co-expression of *MCD1* should partially rescue the deleterious effects on *eco1Δ rad61Δ* cells produced by high-copy *FKH1* or *FKH2* constructs. To test this prediction, *eco1Δ rad61Δ* cells were co-transformed with the various plasmid combinations, serially diluted, and plated onto selective media. As before, high-copy *MCD1* alone rescued *eco1Δ rad61Δ* cell ts-growth defects while high-copy *FHK1* and *FKH2* constructs each exacerbated *eco1Δ rad61Δ* cell ts-growth defects (Fig. 9). In contrast, *eco1Δ rad61Δ* cells that co-overexpressed both *MCD1* and either *FKH1* or *FKH2* exhibited improved growth, relative to either *FKH* alone, even at 37°C (Fig. 9). In combination, these results suggest that Fkh1 and FKh2 both regulate *MCD1* expression in a manner that is highly sensitive to changes in FKH protein levels.

## Swi6 inhibits Mbp1-based transcription of *MCD1*

In yeast, as with most cells, G1 is largely defined as low CDK activity. The rise of G1-CDKs promotes START, a point at which cells commit to cell cycle progression -beginning with transitioning from G1 to S phase. A key facet of START in yeast is the deployment of transcriptional programs required, for instance, for the upregulation of bud growth and DNA synthesis genes (Horak *et al.* 2002). The Mlu I binding factor (MBF) transcription complex

(Swi6 and Mbp1) is one component of START that relies on G1-CDK activity (Costanzo *et al.* 2004; De Bruin *et al.* 2004; Charvin *et al.* 2010). It is therefore intriguing that it is the deletion of the G1 cyclin, *CLN2*, that rescues both *eco1Δ rad61Δ* and *pds5Δ elg1Δ* cell growth defects (Choudhary *et al.* 2022). Previously, *CLN2* deletion was posited to hyperactivate the MBF complex (Swi6, Mbp1) and that MBF may regulate *MCD1* expression (Choudhary *et al.* 2022). These latter two mechanisms, however, remain untested. If these models are correct, then simply overexpressing Swi6 (an activating component of MBF) (Nasmyth and Dirick 1991; Lowndes *et al.* 1992; Morgan *et al.* 1996; Watanabe *et al.* 2011), should rescue the temperature sensitivity of *eco1Δ rad61Δ* cells. To test this possibility, *SWI6* was inserted into a high copy (2μ *TRP1*) plasmid and transformed into both wildtype and *eco1Δ rad61Δ* cells. Log phase wildtype and *eco1Δ rad61Δ* double mutant cells, transformed with either vector alone or vector directing the elevated expression of *SWI6*, were serially diluted, plated onto selective media and incubated at either 30°C or 37°C. Unexpectedly, the overexpression of *SWI6* not only failed to revert the temperature sensitivity of *eco1Δ rad61Δ* cells but, instead, slightly reduced growth at both 30°C and 37°C (Fig. 10a). This adverse effect appears cohesin-dependent given that

elevated *SWI6* expression produced no impact on wildtype cell growth (Fig. 10a).

Swi6 typically is considered to promote Mbp1 activity—the sequence-specific DNA binding component of the MBF transcription complex (Primig *et al.* 1992; Koch *et al.* 1993; Breeden 1996). Other studies, however, suggest that MBF (via Swi6) has an inhibitory role in the transcription of some genes (Dirick *et al.* 1992; Lowndes *et al.* 1992; Sedgwick *et al.* 1998; Li *et al.* 2005; Travesa *et al.* 2013). Together, these findings raised the possibilities that either *MCD1* expression occurs independent of MBF or that Swi6 inhibits a positive role for Mbp1 in *MCD1* expression. To differentiate between these models, *MBP1* was inserted into a high copy (2μ *LEU2*) plasmid. Wildtype and *eco1Δ rad61Δ* cells were then transformed with vector alone, vector driving *MBP1* overexpression, or co-transformed with individual vectors driving *SWI6* and *MBP1* overexpression. *MBP1* overexpression succeeded in partly suppressing the temperature-sensitive growth defects of the *eco1Δ rad61Δ* cells (Fig. 10b). Surprisingly, however, simultaneous overexpression of *SWI6* and *MBP1* not only failed to rescue *eco1Δ rad61Δ* cell ts-growth defects, but fully abrogated any positive growth effect observed by cells expressing *MBP1* alone (Fig. 10c). These findings suggest that Swi6 is a negative regulator of MBF during Mbp1-mediated transcription of *MCD1*.

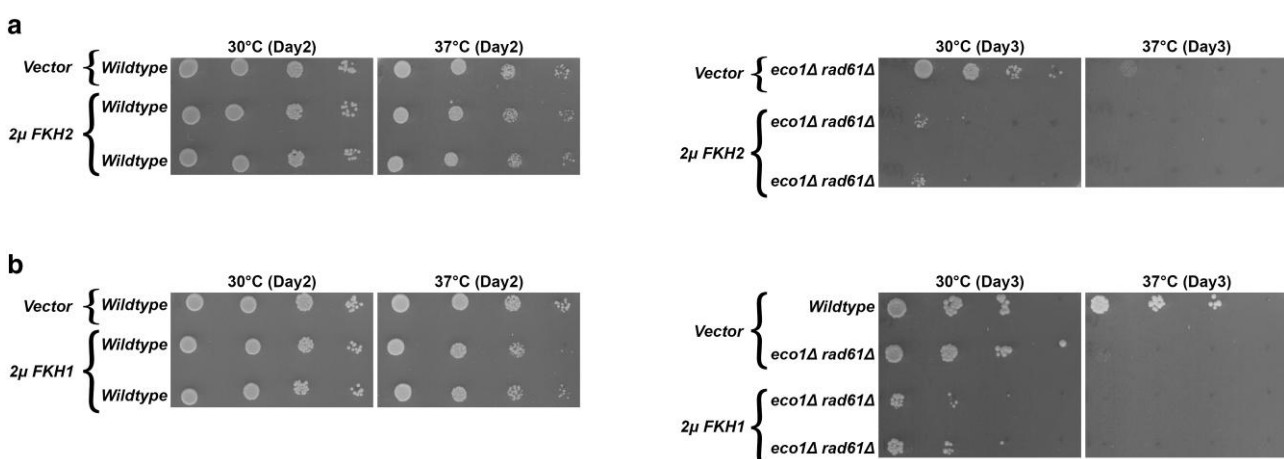

**Fig. 8.** Overexpression of *FKH1* and *FKH2* each are detrimental to the growth of the *eco1Δ rad61Δ* double mutant cells. a) Left panel: Growth of 10-fold serial dilutions of wildtype cells overexpressing vector alone (YGS26), and two independent isolates of wildtype cells overexpressing *FKH2* (YGS126, YGS127). Temperatures and days of growth are indicated. Right panel: Growth of 10-fold serial dilutions of *eco1Δ rad61Δ* double mutant cells overexpressing vector alone (YGS28), and two independent isolates of *eco1Δ rad61Δ* double mutant cells overexpressing *FKH2* (YGS53, YGS55). Temperatures and days of growth are indicated. b) Left panel: Growth of 10-fold serial dilutions of wildtype cells overexpressing vector alone (YGS26), and two independent isolates of wildtype cells overexpressing *FKH1* (YGS79, YGS83). Temperatures and days of growth are indicated. Right panel: Growth of 10-fold serial dilutions of wildtype cells overexpressing vector alone (YGS26), *eco1Δ rad61Δ* double mutant cells overexpressing vector alone (YGS28), and two independent isolates of *eco1Δ rad61Δ* double mutant cells overexpressing *FKH1* (YGS80, YGS84). Temperatures and days of growth are indicated.

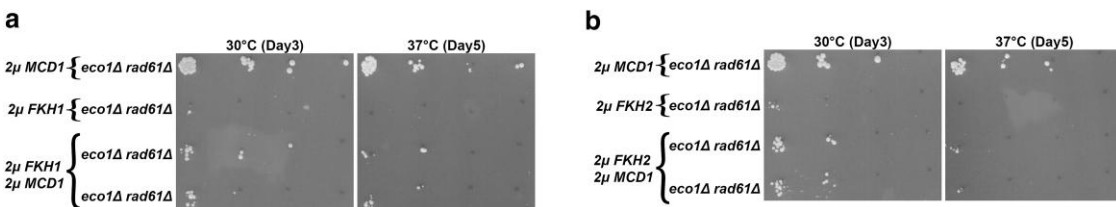

**Fig. 9.** *MCD1* overexpression rescues the deleterious growth effect that result from elevated *FKH1* and *FKH2* levels in *eco1 rad61* null cells. a) Growth of 10-fold serial dilutions of *eco1Δ rad61Δ* doble mutant cells (YMM828) overexpressing *MCD1* (YGS179), *eco1Δ rad61Δ* double mutant cells overexpressing *FKH1* (YGS180), and two independent isolates of *eco1Δ rad61Δ* co-overexpressing *FKH1* and *MCD1* (YGS181, YGS182). Temperature and days of growth are indicated. b) Growth of 10-fold serial dilutions of *eco1Δ rad61Δ* doble mutant cells (YMM828) overexpressing *MCD1* (YGS179), *eco1Δ rad61Δ* double mutant cells overexpressing *FKH2* (YGS183), and two independent isolates of *eco1Δ rad61Δ* co-overexpressing *FKH2* and *MCD1* (YGS184, YGS185). Temperature and days of growth are indicated.

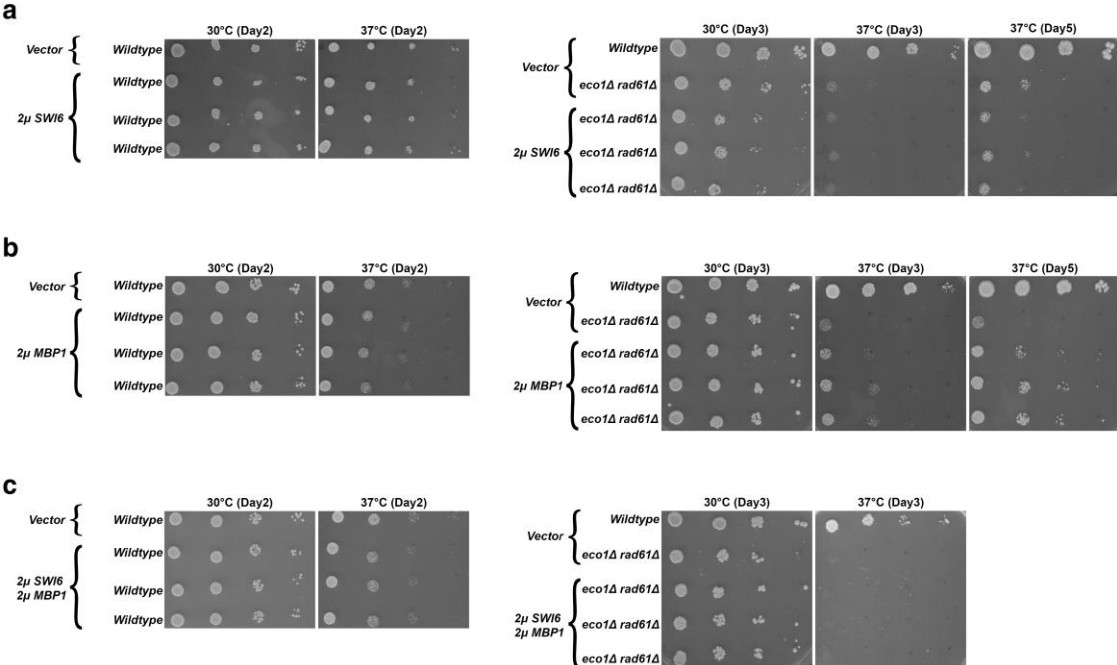

**Fig. 10.** *SWI6* overexpression reduces *eco1 rad61* cell growth, whereas *MBP1* overexpression partially restores *eco1 rad61* cell growth at 37°C. a) Left panel: Growth of 10-fold serial dilutions of wildtype cells overexpressing vector alone (YGS26), and three independent isolates of wildtype cells overexpressing *SWI6* (YGS104, YGS106, YGS108). Temperatures and days of growth are indicated in this and all subsequent panels. Right panel: Growth of 10-fold serial dilutions (dilution series used in all subsequent panels) of wildtype cells overexpressing vector alone (YGS26), *eco1*Δ *rad61*Δ double mutant cells overexpressing vector alone (YGS28), and three independent isolates of *eco1*Δ *rad61*Δ double mutant cells overexpressing *SWI6* (YGS105, YGS107, YGS109). b) Left panel: Serial dilutions of wildtype cells overexpressing vector alone (YGS158), and three independent isolates of wildtype cells overexpressing *MBP1* (YGS162, YGS164, YGS166) are shown. Right panel: Growth of serial dilutions of wildtype cells overexpressing vector alone (YGS158), *eco1*Δ *rad61*Δ double mutant cells overexpressing vector alone (YGS160), and three independent isolates of *eco1*Δ *rad61*Δ double mutant cells overexpressing *MBP1* (YGS163, YGS165, YGS167). Temperatures and days of growth are indicated. c) Left panel: Serial dilutions of wildtype cells overexpressing two empty vectors (YGS168), and three independent isolates of wildtype cells overexpressing *SWI6* and *MBP1* (YGS170, YGS171, YGS172) are shown. Right panel: Serial dilutions of wildtype cells overexpressing two empty vectors (YGS168), *eco1*Δ *rad61*Δ double mutant cells overexpressing two empty vectors (YGS130), and three independent isolates of *eco1*Δ *rad61*Δ double mutant cells overexpressing *SWI6* and *MBP1* (YGS134, YGS138, YGS142).

We considered the possibility that Swi6, in addition to inhibiting Mbp1-dependent expression of *MCD1*, could be deregulating other genes that exacerbate *eco1*Δ *rad61*Δ cell viability. If correct, this model predicts that co-expressing *MCD1* along with *SWI6* should fail to rescue *eco1*Δ *rad61*Δ cell ts-growth defects. In contrast, a partial rescue would indicate that the critical role of Swi6 centers on *MCD1* repression. To differentiate between these predications, *eco1*Δ *rad61*Δ cells were transformed with vector driving overexpression of *MCD1* and *SWI6* individually, as well as co-transformed with both *SWI6* and *MCD1* vectors. The resulting transformants were serially diluted, plated onto selective media and incubated at either 30°C or 37°C. As expected, *eco1*Δ *rad61*Δ cells overexpressing *SWI6* exhibited diminished growth while cells overexpressing *MCD1* exhibited improved growth 30°C (Fig. 11). Importantly, the slow growth phenotype exhibited by *eco1*Δ *rad61*Δ cells that contained *SWI6* alone was fully rescued by co-expression of *MCD1* at 30°C (Fig. 11). Moreover, while *eco1*Δ *rad61*Δ cells overexpressing *SWI6* were inviable at 37°C, co-expression of *MCD1* largely rescued cell viability at 37°C. These findings provide strong evidence that Swi6 is a negative regulator of Mbp1-mediated transcription of *MCD1*.

## Discussion

Conservatively, cohesins could be considered master regulators of DNA in that cohesin mutations impact replication origin firing, nuclear organization, gene transcription, sister chromatid

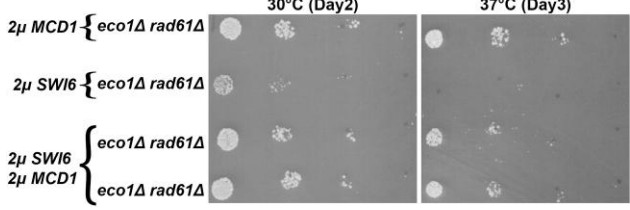

**Fig. 11.** Swi6 negatively regulates Mbp1-based transcription of *MCD1*. Growth of 10-fold serial dilutions of *eco1*Δ *rad61*Δ double mutant cells (YMM828) overexpressing *MCD1* (YGS179), overexpressing *SWI6* (YGS197), and two independent isolates co-overexpressing *SWI6* and *MCD1* (YGS199, YGS200). Temperature and days of growth are indicated.

segregation, chromosome condensation, and high-fidelity DNA repair (Guacci *et al.* 1997; Michaelis *et al.* 1997; Uhlmann and Nasmyth 1998; Skibbens *et al.* 1999; Zhang *et al.* 2008; Kulemzina *et al.* 2012; Tong and Skibbens 2014, 2015; Eng *et al.* 2015; Cattoglio *et al.* 2019). Often, factors and processes of such wide-ranging importance are placed under surveillance mechanisms (Vaddavalli and Schumacher 2022; Kalmykova 2023; Kornepati *et al.* 2023; Maiato and Silva 2023). While cohesins have been extensively studied, whether cells monitor cohesin integrity remains largely unknown. The first revelation of this study involves the potential emergence of a new class of checkpoint. Mcd1 must be degraded (to allow for sister chromatid segregation during mitosis) and then replenished (induced transcription at the

G1/S transition in yeast cells) in every cell cycle. Here, we report that Mcd1 levels are precipitously reduced in both *eco1Δ rad61Δ* cells. Prior studies similarly note reductions in Mcd1 levels in *pds5Δ elg1Δ* cells as well as cells mutated for either *SMC1* or *SMC3* or *RAD61* (Toth *et al.* 1999; Bloom *et al.* 2018; Choudhary *et al.* 2022; Boardman *et al.* 2023). We thus posit that regulating Mcd1 levels provide for a novel "protection" mechanism akin to apoptosis: that defects in cohesin structure promote elevated and/or prolonged Mcd1 degradation. However, this can only be half of the "checkpoint," given that *MCD1* is newly transcribed each and every cell cycle. Importantly, *MCD1* levels apparently fail to rise in any of the mutated cells listed above. This suggests a feedback mechanism in which defects in cohesin structure also block *MCD1* transcription. In this way, cohesin defects invoke transcriptional control over *MCD1* (and possibly degradative control over Mcd1) to preclude the rise of an aneuploid population. It is tempting to speculate that cohesins of the prior cell cycle form a positive feedback loop by impacting the transcriptional networks required for subsequent *MCD1* expression. Our discovery that elevating Mcd1 levels is solely sufficient to rescue *eco1Δ rad61Δ* cell inviability at 37°C, and similarly rescues other cohesin-mutated cell inviabilities (Bloom *et al.* 2018; Choudhary *et al.* 2022), suggests that an *MCD1*-dependent checkpoint pathway may monitor for a wide range of cohesin defects. Future efforts will be required to test the extent to which RAD21 (human homolog of Mcd1) culling may protect multicellular organisms from tumorigenesis and developmental abnormalities.

By extension, central questions now emerge regarding the fundamental mechanism(s) that regulate *MCD1* expression—control of which likely underlies all subsequent cohesin functions. In this light, the second revelation of current study is the identification of multiple transcriptional pathways that appear to regulate *MCD1* (Fig. 12). Fdo1 is the first such regulator of *MCD1* expression in that *FDO1* deletion rescues *eco1Δ rad61Δ* cell ts-growth and that

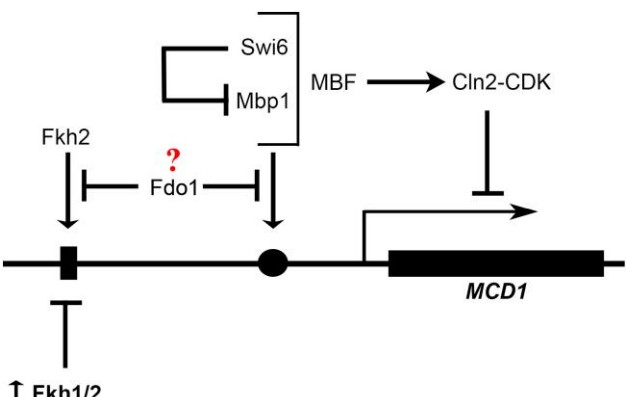

**Fig. 12.** Schematic highlights integrated mechanisms of *MCD1* regulation. MBF complex (Swi6 and Mbp1) and Forkhead (Fkh1 and Fkh2) transcription factors bind non-overlapping DNA sequences upstream of *MCD1*. Swi6 and Mbp1 play antagonistic roles in *MCD1* expression: Swi6 inhibits the positive role that Mpb1 otherwise performs in *MCD1* transcription. Of the two forkhead transcription factors in yeast, Fkh2 appears to play a positive role in *MCD1* expression, although elevated expression of either *FKH1* or *FKH2* (in bold with up arrow) is detrimental to *eco1Δ rad61Δ* cell growth. Fdo1, a co-repressor that binds Fkh1, inhibits *MCD1* expression. Notably, elevated expression of *MCD1* rescues (or partially suppresses) the negative effects produced by alterations in any one of these three (MBF, FKHs, and Fdo1) pathways. Further analysis is required to clarify the dual role of FKHs in regulating *MCD1* expression, as well the molecular mechanisms through which Cln2-CDK may impact the transcriptional pathways defined here to regulate Mcd1 levels.

*MCD1* elevated expression rescues *eco1Δ rad61Δ* cell inviability that otherwise arises in response to *FDO1* elevated expression. As a direct-binding transcription factor repressor, Fdo1 is unique compared to other gene deletions (*RAD61*, *ELG1*, and *CLN2*) that can rescue cohesin mutant cell inviability. Rad61 (an auxiliary cohesin subunit) and Elg1 (regulating PCNA-dependent recruitment of Eco1 to the DNA replication fork) directly impact cohesin function (Skibbens *et al.* 1999; Kueng *et al.* 2006; Moldovan *et al.* 2006; Ben-Shahar *et al.* 2008; Heidinger-Pauli *et al.* 2009; Maradeo and Skibbens 2009, 2010; Parnas *et al.* 2009; Rowland *et al.* 2009; Sutani *et al.* 2009; Bender *et al.* 2020; Zuilkoski and Skibbens 2020; Buskirk and Skibbens 2022; Choudhary *et al.* 2022). Cln2 (G1 cyclin component of CDK) appears to likely regulate *MCD1* expression indirectly through post-translational modifications. *FDO1* deletion is also unique to other allele-specific mutations that encode cohesin subunits (Irr1/Scc3 and Pds5) or cohesin modifiers (Eco1) that also can rescue *eco1Δ* cell inviability (Rowland *et al.* 2009; Sutani *et al.* 2009). This latter category is comprised of mutations in essential factors that normally are required for cohesin function and therefore are not negative regulators in the strictest sense in which full gene deletion rescues Eco1-deficient cell inviability.

Related discoveries include the extended and complex roles of additional transcriptional networks that overlay *MCD1* expression (Fig. 12). Fkh1 and the Fkh2 transcriptional factor paralogs were previously reported to regulate mating-type switching (silencing), replication origin firing (Casey *et al.* 2008), RNA Polymerase II transcription elongation (Morillon *et al.* 2003), and transcription of the *CLB2*-cluster genes (Koranda *et al.* 2000; Kumar *et al.* 2000). Here, we report that *eco1Δ rad61Δ* cells are highly sensitive to increased Fkh1 and Fkh2 levels and that the resulting exacerbated growth in each case is ameliorated by co-expression of *MCD1*. Fkh1 and Fkh2 differ however in that only deletion of *FKH2* adversely impacts *eco1Δ rad61Δ* cell growth—an effect also reversed by elevated expression *MCD1*. Intriguingly, biochemical studies found that Fdo1 interacts with the FOX-type transcription factor Forkhead 1 (Fkh1) (Ho *et al.* 2002; Dummer *et al.* 2016). Given the antagonistic and dose-dependent effects reported here, future efforts may be challenging given that Forkhead transcription factors, which are widespread from yeast to humans, bind over 100 proteins (via forkhead binding-domains) that play critical roles in development and ciliogensis, metabolism, ageing and cancer (Jonsson and Peng 2005; Jin *et al.* 2020; Lewis and Stracker 2021).

The final revelation of the current study centers on our analyses of the MBF transcription complex—including the antagonistic activities of each MBF component on *MCD1* expression (Fig. 12). A prior study inferred the role of MBF in promoting *MCD1* expression, but this model remained untested (Choudhary *et al.* 2022). MBF is comprised of Swi6 and Mbp1. An analogous transcriptional regulator of START, SBF, is comprised of Swi6 and Swi4 (Primig *et al.* 1992; Moll *et al.* 1993; Dirick *et al.* 1995). In both cases, Swi6 is positioned as a positive regulator of Mbp1 and Swi4, the latter two factors providing DNA sequence-dependent promoter binding activities. Our findings suggest that Swi6 is a strong negative regulator of the positive role that Mbp1 performs in *MCD1* expression, consistent with results from a high-through put study that *SWI6* deletion promotes *mcd1* mutant cell growth (128). These results also are consistent with prior studies that (1) MBF/SBF transcription continues in the absence of Swi6, although the expression of those genes are no longer correctly regulated in the cell cycle (Dirick *et al.* 1992; Lowndes *et al.* 1992; Li *et al.* 2005). In fact, prior studies revealed that SBF and MBF can act as transcriptional suppressors such that loss of *SWI4* or *MBF*

enhances the transcription of certain genes (Bean *et al.* 2005). Clearly, the roles of MBF and SBF are nuanced and results, such as these involving *MCD1*, must be viewed through the lens of specific gene expressions.

## Data availability

Strains and plasmids are available upon request. The authors affirm that all data necessary for confirming the conclusions of the article are present within the article, figures, and tables with the following exceptions. Supplementary Table 1 contains yeast strains and genotypes used in this study. Supplementary Table 2 contains DNA oligo sequences used in this study. Supplementary Figs. 1–4 contain supplemental figures and figure legends. Illumina sequencing data was deposited into the short read archive (SRA): BioSample SAMN42761258 within the BioProject PRJNA836598.

Supplemental material available at GENETICS online.

## Acknowledgements

The authors thank past and present Skibbens lab members (Annie Sanchez, Grace Duke, Abbie Brown, Fiona Mensching, Niusha Banoukh, and MJ Schwab) for helpful discussions during the preparation of this paper. The authors gratefully acknowledge the generous gift of Mcd1-directed antibody by Dr. Vincent Guacci and expertise in genome sequence analyses performed by Prof. Sean Buskirk.

## Conflicts of interest

The author(s) declare no conflict of interest.

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

*Editor: N. Bhalla*