## [Peer Review File · Genetics]

Fdo1, Fkh1, Fkh2 and the Swi6-Mbp1 MBF complex regulate Mcd1 levels to impact *eco1 rad61* cell growth in *Saccharomyces cerevisiae*

Gurvir Singh and Robert Skibbens

NOTE: The reviews and decision letters are unedited and appear as submitted by the reviewers.

In extremely rare instances and as determined by a Senior Editor or the EIC, portions of a review may be redacted. If a review is signed, the reviewer has agreed to no longer remain anonymous.

The review history appears in chronological order.

Review Timeline:

Submission Date:	2024-02-14
Editorial Decision:	2024-03-08
Resubmission Received:	2024-06-10
Editorial Decision:	2024-07-09
Revision Received:	2024-07-16
Accepted:	2024-07-19

March 8, 2024

GENETICS-2024-306872

FDO1 deletion and Mbp1 activation elevate Mcd1 levels to rescue *eco1 rad61* cell growth defects in *Saccharomyces cerevisiae*

Dear Dr. Skibbens:

Two experts in the field have reviewed your manuscript, and I have read it as well. We all appreciated the potential insight into the regulatory pathways that control cohesion expression. While your manuscript is not currently acceptable for publication in GENETICS, we would welcome a substantially revised manuscript. Both reviewers have comments and concerns to be addressed in a revised manuscript. You can read their reviews at the end of this email.

Both reviewers highlighted that some claims made in the paper needed additional experiments, such as looking at Mcd1p levels, to support them, provided suggestions for how to better quantify and present some of the data in Figures 4B and C and asked for revisions to the manuscript so that it is more broadly accessible. We look forward to receiving your revised manuscript. Please let the editorial office know approximately how long you expect to need for revisions.

Upon resubmission, please include:

1. A clean version of your manuscript;
2. A marked version of your manuscript in which you highlight significant revisions carried out in response to the major points raised by the editor/reviewers (track changes is acceptable if preferred);
3. A detailed response to the editor's/reviewers' feedback and to the concerns listed above. Please reference line numbers in this response to aid the editor and reviewers.

Your paper will likely be sent back out for review.

Additionally, please ensure that your resubmission is formatted for GENETICS
<https://academic.oup.com/genetics/pages/general-instructions>

Follow this link to submit the revised manuscript: Link Not Available

Sincerely,

Needhi Bhalla
Associate Editor
GENETICS

Approved by:
Jeff Sekelsky
Senior Editor
GENETICS

Reviewer #1 (Comments for the Authors (Required)):

Cohesin complexes play a plethora of functions throughout cell cycles, but the regulation of cohesin complexes remains not well understood. In this manuscript, the authors report FDO1 as a novel regulator of cohesins, which suppresses the temperature-sensitive phenotype of the *eco1Δ rad61Δ* double mutant. They identified FDO1 from a suppressor screen and confirmed its identity by mutating the FDO1 gene in the *eco1Δ rad61Δ* double mutant background, showing that *fdo1Δ* allowed the cells to grow at 37°C. Next, the authors examined whether interactors of Fdo1, the FOX-type transcription factors Fkh1 and Fkh2, lead to a similar growth promotion phenotype at a non-restrictive temperature when mutated. They found that while neither *fkh1Δ* nor *fkh2Δ* rescued the lack of growth phenotype, *fkh2Δ* showed more severe growth defects. To test the possibility that Fdo1 inhibited the activation of Fkh1 and Fkh2, the authors overexpressed Fkh1 and Fkh2 in the *eco1Δ rad61Δ* double mutant and found that Fkh1 and Fkh2 negatively impacted cohesin pathways, as their overexpression led to growth defects even at 30°C. The authors then measured Mcd1 levels in S phase-arrested cells and showed that it was reduced in the *eco1Δ rad61Δ* double mutant compared to the wild type but elevated when Fdo1 was mutated. By overexpressing Mcd1, the authors beautifully demonstrated that it was sufficient to rescue growth at 37°C in the double mutant, providing evidence that Fdo1 deletion led to an increase in Mcd1 level, thereby rescuing the temperature sensitivity of the *eco1Δ rad61Δ* double mutant. The authors also overexpressed Swi6 in the double mutant, but this overexpression did not promote growth at

37°C. However, overexpressing Mbp1 partially reverted the phenotype at 37°C. In contrast, overexpressing Swi6 and Mbp1 at the same time did not yield any positive effect on growth promotion at 37°C, suggesting that Swi6 functions as a negative regulator of MBF.

The authors did a great job of providing a well-organized and clearly written manuscript with an easy-to-follow logic flow. They also provided strong evidence to support their conclusions about the roles of Fdo1 and Mbp1 in regulating Mcd1, which is essential in promoting growth at the restrictive temperature in the *eco1Δ rad61Δ* double mutant. Overall, this is a great piece of work that enhances our understanding of the subject of cohesin regulation, and it will be of interest to the audience of Genetics. I have a few comments listed below:

1. I would like the authors to provide a short discussion about what they think the role of Mcd1 involves in rescuing the growth of *eco1Δ rad61Δ* double mutant.
2. Line 98: Could you elaborate on NIPBL? The transition from the previous sentence about CdLS patients is a bit abrupt for readers not familiar with this area.
3. Line 122: "cohesin dissociation" instead of "cohesion."
4. Line 148: Three independent *eco1Δ rad61Δ fdo1Δ* triple mutants were mentioned, but the previous sentence from lines 145-147 indicates that there were only two, one with the complete coding sequence removed and the other had the same partial deletion as the one from the suppressor screen. What is the last one?
5. Lines 149 and 152: *fdo1Δ* instead of *fdo1D*.
6. Figure 1A and B: Maybe label isolate 1, 2, 3, etc.?
7. Line 153: Why did you decide to examine *fdo1C247T*? What is special about the C terminus of *fdo1*? How long is the FDO1 gene, and how many amino acids does it have?
8. Line 153: In the parenthesis, should it be 83-342Δ?
9. Lines 200-201: While I understand your logic, I do think that the first hypothesis "whether elevated expression of FDO1 would rescue *eco1Δ rad61Δ* cell ts growth" sounds a bit weird, as you have shown that deletion of FDO1 leads to reversion of the phenotype, so why would the opposite of deletion lead to the same result?
10. Line 256: "mcd1-1" should be *eco1Δ rad61Δ*?
11. Line 267: Could you elaborate on START, MBF, and this first sentence to give a more detailed background?
12. Line 292: Mbp1 instead of Mpb1.
13. Line 361: consistent "with" results from.

Reviewer #2 (Comments for the Authors (Required)):

The study proposes a pathway in which the mating type switching factor FDO1 regulates expression of the cohesion subunit MCD1, via the MBF complex. The *fdo1* gene was identified as a spontaneous revertant in a suppressor screen in the *eco1 rad61* background. A previous study by the same group described another hit from the screen, *cln2* (Buskirk et al 2022). The proposed pathway is novel and interesting, but the data, as presented in the current manuscript, do not support the conclusions. Additional experiments would be needed to be allowed to make the conclusions as stated.

Major points:

1. Fig 2: Mutation/deletion of *fdo1* rescues the temperature sensitive growth defects of *eco1 rad61* double mutants. However, deletion of *fkh1* or *fkh2*, encoding biochemical interactors of FDO1, does not rescue. Since these two genes are redundant in their cell cycle control role (according to ref 111, and as cited by authors), could they be functioning redundantly in this pathway, too? Does deletion of both *fkh1* and *fkh2* rescue the growth defects?
2. Fig 3: Overexpression of FDO1, FKH1, or FKH2 all exacerbate the growth defects. This finding makes it very difficult to propose a way in which FDO1 and one of the FKH may work together or against each other. The authors propose that these findings mean that FKH1 and FKH2 negatively impact cohesion pathways (lines 197-198). I disagree with this conclusion. The lethality could be due to some other function entirely unrelated to cohesin. An experiment to support the claim that FKH1 and 2 regulate cohesion would be to check MCD1 protein levels (similar to the experiments in Fig 4).
3. Fig 4: The authors argue that protein levels of the cohesion subunit MCD1 decrease in *eco1 rad61* mutants, but that this defect is partially rescued by deletion of *fdo1*. The data presented does not support this conclusion, at least not the second part, that MCD1 protein levels are higher in *eco1 rad61 fdo1* triple mutants compared to *eco1 rad61* double mutants. The difference is not statistically significant. The error bars, representing SEM, are huge, indicating the variability between replicates must be significant. Seeing the raw numbers (in a supplemental statistical table) might help evaluate the data better. If the authors wish to make the conclusion that the triple mutant has higher protein levels than the double, the differences need to be statistically significant. Ideally, the experiment should be done using quantitative westerns. If not possible, at the very least the authors should do a dilution series for the proteins to demonstrate that they are in the linear range of detection. One of these methods may help reduce the variability between samples.
4. Fig 5: the data presented on this figure shows that MCD1 overexpression can in fact rescue the ts growth defects. If the argument is to be made that *fdo1* deletion rescue due to an increase in MCD1 levels, the authors should compare MCD1 protein levels in *eco1 rad61 fdo1* triple mutants to MCD1 overexpression in the double (on a quantitative western) and compare the relative protein levels to the effects on viability.

5. The Swi6 and Mbp1 results are again confusing. It again would help to see MCD1 protein levels in SWI6 and MBP1 overexpression lines to support the authors argument about the roles of these proteins in promoting MCD1 expression.
6. A final connection the authors try to make is that FDO1 represses mbp1 (and therefore its deletion leads to MBP1 overexpression and rescue). To support this argument, they would need to test whether deletion of both fdo1 and mbp1 (in an eco1 rad61 fdo1 mbp1 quadruple mutant) abrogates the rescue seen in eco1 rad61 fdo1 triple mutants.

Minor points:

1. The writing needs work. There are several grammatical errors, and examples of improper choice of words. For example, the word "refractile" in line 259 is probably not the word the authors meant to use.
2. Background information could be presented better. For example, the abstract should state that MCD1 is a cohesion subunit. Line 98 should explain what NIPBL is.
3. Fig 8: Panel A is not needed. The authors do not do any experiments on the cln2 pathway. Panel B summarizes the model they propose. However, as explained above, additional experiments would be needed to actually propose this model.

Reviewer #3 (Comments for the Authors (Required)):

Cohesin is a four-subunit complex (Smc1p, Smc3p, Mcd1p, Scc3p) that mediates multiple chromosome structure and function aspects. These include sister chromatid cohesion, condensation, DNA damage repair, transcriptional regulation, and chromosome segregation. Two cohesin regulators, Eco1p and Rad61p perform antagonistic functions. Eco1p acetylates Smc3p to stabilize cohesin binding to chromosomes, which enables the establishment of sister chromatid cohesion, whereas Rad61p destabilizes the chromosomal binding of non-acetylated cohesin. Given that only ~20% of cohesin is acetylated, a large pool of cohesin can be utilized for cohesin functions other than sister chromatid cohesion. All four cohesin sub-units are essential genes as is ECO1. In budding yeast, the deletion of RAD61 restores viability to eco1Δ mutants. However, the eco1Δ rad61Δ double mutant (eco1Δ rad61Δ) cells have numerous defects including highly defective cohesion, drug sensitivity and temperature sensitivity. Significant new insights on cohesin regulation have come from identifying and characterizing suppressors of these deficiencies of the eco1Δ rad61Δ. This paper describes a new set of suppressors that have the potential to provide important insights into cohesin regulation.

The authors had previously conducted a genetic screen to identify suppressors of the eco1Δ rad61Δ temperature sensitivity and identified cln2Δ as a suppressor. Here, the authors identify and characterize an fdo1 mutant as another such suppressor. Over-expression of FDO1 was toxic to eco1Δ rad61Δ cells, solidifying a genetic relationship. The authors examined Mcd1p levels and showed that it was reduced in eco1Δ rad61Δ cells, but this reduction was lessened in the triple mutant (fdo1Δ eco1Δ rad61Δ), which provides evidence that suppression is due to increased Mcd1p levels. This conclusion is solidified by experiments showing that Mcd1p over-expression using a high copy plasmid bearing MCD1 provides an even more robust suppression than fdo1Δ does. They expand the genetic pathways by showing that over-expression of the transcriptional regulator MBP1 and its partner have opposite effects. MBP1 over-expression suppresses the eco1Δ rad61Δ temperature sensitivity, whereas SWI6 over-expression exacerbates the temperature sensitivity. This divergence suggests that MBP1 and SWI6 have opposing actions with regard to controlling MCD1 expression. The authors present a model to explain suppression. They suggest that several transcription pathways control Mcd1p levels which can be altered to increase Mcd1p levels and promote suppression.

The experiments are generally well-controlled, and conclusions from most are consistent with the authors' interpretations. The data supports their model that MCD1 expression is under the control of different regulatory mechanisms and that perturbing this regulation to increase MCD1 expression suppresses the eco1Δ rad61Δ temperature sensitivity. However, the paper seems light in the amount of data presented, and the authors fail to present biological data to understand why increasing Mcd1p levels suppresses temperature sensitivity. As such, the paper is not yet suitable for publication in Genetics. A few simple experiments should be done to correct these deficiencies. First, the authors should test their model more directly by over-expressing MCD1 in eco1Δ rad61Δ cells deleted for either fkh2Δ or fkh1Δ. The same goes for over-expressing MCD1 in eco1Δ rad61Δ cells also over-expressing SWI6 or one of the FKH1/2 or FDO1. The model would be strengthened if MCD1 over-expression does indeed suppress the temperature sensitivity or counteract the deleterious effects. It may be necessary to place MCD1 under control of an inducible promoter for such analyses to avoid any dominant inhibition of the native MCD1 promoter. It may be wise to add an intermediate temperature between 30{degree sign}C and 37{degree sign}C to detect more subtle effects. A bigger deficiency is that the authors make no attempt to assess the biological basis for suppressing the eco1Δ rad61Δ temperature sensitivity. eco1Δ rad61Δ cells have defects such as severe cohesion defects, delays in the cell-cycle progression, as well as extreme sensitivity to drugs that damage DNA or destabilize microtubules. The paper would be greatly strengthened by comparing the phenotypes of the double mutant and the double mutant with suppressors (minimally fdo1Δ and MCD1 overexpression) by simple established tests for cohesion, drug sensitivity, and cell cycle progression. These data would inform on which cohesin functions are rate limited by the levels of Mcd1 activity.

INTRODUCTION:

The introduction reads more like a review than introduction to a Genetics paper dealing with suppression of the temperature sensitivity of eco1Δ rad61Δ cells. Too much detail and discussion of topics like TADs and looping when nothing in the results

deals with TADs or looping. The entire 4th paragraph is about Roberts syndrome and end point they make is that cohesin functions not related to sister chromatid cohesion are important. The focus should be on the latter, especially data from *eco1Δ rad61Δ* double mutants, but Roberts syndrome can be briefly mentioned as one example of the importance of non-cohesin functions in human cells. There is basically no mention of phenotypes of *eco1Δ rad61Δ* mutants other than temperature sensitivity. The last important omission is that there is no mention that cohesin subunit levels fluctuate during the cell-cycle with expression increased in S phase. More importantly, Mcd1p is destroyed in anaphase and G1 phase so little remains in cells at these stages, making Mcd1p different from other cohesin subunits. This difference suggests that proper regulation of Mcd1p levels is of prime importance for cohesin function. Introducing the unique regulation of Mcd1p levels sets the stage for explaining their suppression of the temperature sensitivity of *eco1Δ rad61Δ* double mutants.

RESULTS

General for Figures:

A. It would be easier for the reader if the labels for 2 plasmids were placed on the same side as the genotypes rather than on the other side of the images.

B. The model presented discusses regulation of MCD1 gene expression. It would greatly strengthen the model if some RT-qPCR was used to quantify steady-state levels of MCD1 RNA levels, at least for conditions with the strongest effects.

Figure 4A. The FACS profiles are mis-labelled. The peaks should be labelled 1C and 2C for haploid genomes before and after DNA replication. A label of 1N and 2N means cells are a mix of haploids and diploids.

Figure 4B. The blots are problematic. There are two points to be made in these blots and they are best examined separately not in one blot. First, not all antibodies yield a linear signal. The authors should do a dilution series to avoid such issues. In the first blot they show Mcd1p levels are reduced in *eco1Δ rad61Δ* cells. Use WT at 100%, 75%, 50% and 25% and compare to 100% of *eco1Δ rad61Δ* cell extracts or some dilution of it. A second blot should be focused solely on a comparison of a dilution series of *fdo1Δ eco1Δ rad61Δ* extracts to *eco1Δ rad61Δ*. After all, the key point is how does *fdo1Δ* affect Mcd1p levels in *eco1Δ rad61Δ* cells. That is the reference, not how Mcd1p levels in *fdo1Δ eco1Δ rad61Δ* compares to WT cells.

The extracts themselves seem a bit problematic. The authors claim band labelled by a star is a non-specific band. This seems unlikely as a non-specific band should be the same in all extracts but it is stronger in WT and the triple mutant. It appears to be an Mcd1p degradation product as are other bands with a slower mobility. The authors used a zymolyase treatment to generate protein extracts. That is dangerous as Mcd1p is easily degraded by several proteases. They should prepare new extracts using a TCA proposal to prevent degradation.

DISCUSSION:

Line 312-313. It isn't precise to group FDO1, RAD61, CLN2 and ELG1 as negative regulators of cohesin. FDO1 and CLN2 negative regulators of cohesin transcription whereas RAD61 and ELG1 can be considered negative regulators of cohesin function.

Line 317-320. This is a bit general as well as inaccurate. Like *rad61Δ*, certain alleles of SCC3 and PDS5 can also suppress the inviability of *eco1Δ*, and these double mutants also have dramatic cohesion defects and drug sensitivities. ECO1 is not a cohesin subunit but Line 319 is written to suggest it is.

Missing from the discussion: It has been shown that *rad61Δ* (i.e. *wpl1Δ*) has ~a 50% reduction in Mcd1p levels and is bound to chromosomes, yet cells are not temperature-sensitive. It is important to mention this distinction early: Reduced Mcd1p levels alone don't cause temperature sensitivity. Rather, the Ts-phenotype only occurs in the absence of Eco1p in a *rad61Δ* background.

Associate Editor Comments:

Dear Prof Bhalla,

We thank the reviewers and you, our Editor, for their time and careful consideration in reviewing our submitted work. We also greatly appreciate the overall positive nature of their comments: "*strong evidence to support their conclusions*" from reviewer 1, the "*proposed pathway is novel and interesting*" from reviewer 2, and that "*Significant new insights on cohesin regulation have come from identifying and characterizing suppressors ... that have the potential to provide important insights into cohesin regulation*" from reviewer 3.

While we remain very excited about the mechanisms discovered in this study, we took full note of the reviewers' comments and concerns. The revised study includes several new experiments (including lots of Westerns and also analyses of new strains) and significant manuscript revisions. We trust that you and, at your discretion, our reviewers will find the revised study substantially improved and that our new findings support completely the original conclusions. To briefly highlight a few of these changes/additions:

1. We re-performed all Western blots, after first establishing a linear range of detections that were suitable for quantifications. All results (for instance, Mcd1 levels are reduced in *eco1 rad61* cells and also *FDO1* deletion increases Mcd1 levels) are now statistically significant.
2. The impact of Mcd1 levels are now assessed for each of the transcription pathways (*Fdo1*, *Fkh1*, *Fkh2*, *Swi6*, and *Mbp1*) reported in this study. Each experiment performed confirms that the adverse growth impacts produced by deletion (or elevated expression) of a transcriptional factor can be rescued solely by the elevated expression of *MCD1*.
3. Reviewer 3 wondered if the double deletion of *fkh1* and 2 might be required to rescue *eco1 rad61* ts growth. The effect of double mutants (ie., *fkh1* and *fkh2*) on *eco1 rad61* cells are now reported in this study. The combination is lethal - consistent with our original findings that *fkh2* deletion, at least, greatly exacerbates *eco1 rad61* ts growth.
4. The introduction has been significantly streamlined (2.2 pages down from 3.3 pages).
5. The Results section has been revised to accommodate numerous new experiments and the re-ordering of key sections to enhance the overall flow.
6. A Data Availability Statement is now included in the ms.

Below, we provide detailed responses (embedded in *italics*) to each and every comment provided by our reviewers. Lines refer to the ms marked version. Again - we thank our Editor and reviewers for this opportunity. We feel that the inclusion of these new experiments dramatically strengthened the conclusions of this study.

REVIEWER 1

The authors did a great job of providing a well-organized and clearly written manuscript with an easy-to-follow logic flow. They also provided strong evidence to support their

conclusions about the roles of Fdo1 and Mbp1 in regulating Mcd1, which is essential in promoting growth at the restrictive temperature in the *eco1Δ rad61Δ* double mutant. Overall, this is a great piece of work that enhances our understanding of the subject of cohesin regulation, and it will be of interest to the audience of Genetics.

We very much appreciate the enthusiastic support for the study

I have a few comments listed below:

1. I would like the authors to provide a short discussion about what they think the role of Mcd1 involves in rescuing the growth of *eco1Δ rad61Δ* double mutant.

*We thank the reviewer for the opportunity to expand on this important, and rather intriguing, topic. As the reviewer knows, Mcd1 must be degraded (to allow for sister chromatid segregation during mitosis) and then replenished (induced transcription during G1 in yeast cells) in every cell cycle. We report here that Mcd1 levels are precipitously reduced in *eco1Δ rad61Δ*, expanding on prior results that cells deficient in other cohesin pathways also contain reduced Mcd1 protein levels. In responding to the reviewer's request, we now propose a novel 'protective' mechanism (akin to apoptosis) in which defects in cohesin structure block MCD1 transcription (and likely promote elevated and/or prolonged Mcd1 degradation). The Discussion of the manuscript (lines 1213-1256) briefly introduces this new model that repressed MCD1 transcription, and possibly increased degradation, may represent a type of checkpoint that abrogates cohesin functions to ensure cell death and preclude the persistence of aneuploid cells.*

2. Line 98: Could you elaborate on NIPBL? The transition from the previous sentence about CdLS patients is a bit abrupt for readers not familiar with this area.

In streamlining the Introduction, this portion of the text was deleted from the manuscript.

3. Line 122: "cohesin dissociation" instead of "cohesion."

This portion of text is now deleted.

4. Line 148: Three independent *eco1Δ rad61Δ fdo1Δ* triple mutants were mentioned, but the previous sentence from lines 145-147 indicates that there were only two, one with the complete coding sequence removed and the other had the same partial deletion as the one from the suppressor screen. What is the last one?

We appreciate the opportunity to clarify the results. We generated many independent isolates for both null and truncated Fdo1. The text (lines 307-318) has been revised accordingly to differentiate between the number of isolates tested.

5. Lines 149 and 152: *fdo1Δ* instead of *fdo1D*.

Thanks - corrected (now lines 311 and 314)

6. Figure 1A and B: Maybe label isolate1, 2, 3, etc.?

We revised the text to make clear the number of independent isolates tested. The revised text (now Figure Legend 2, lines 2053-2064) also includes each of the separate strain names.

7. Line 153: Why did you decide to examine fdo1C247T? What is special about the C terminus of fdo1? How long is the FDO1 gene, and how many amino acids does it have?

Thanks for raising this issue - the C247T mutation is the allele identified in our screen. We agree that bouncing back and forth between DNA sequence numbers and amino acid sequence numbers is confusing. We revised the manuscript and figure legends in two ways. First, we deleted references to DNA sequence and now refer only to amino acid sequence. Second, we revised the truncated name to indicate that Fdo1 protein extends only to aa 82, ie to fdo1^{1-82Δ}. Revised text are found in lines 315-318.

8. Line 153: In the parenthesis, should it be 83-342Δ?

Yes - we revised to standard nomenclature (fdo1^{1-82Δ}) as described above.

9. Lines 200-201: While I understand your logic, I do think that the first hypothesis "whether elevated expression of FDO1 would rescue eco1Δ rad61Δ cell ts growth" sounds a bit weird, as you have shown that deletion of FDO1 leads to reversion of the phenotype, so why would the opposite of deletion lead to the same result?

We appreciate the reviewer's comment that the guiding principle of the experiment was sound, but agree that the logic (based on fkh deletion results) was overly complicated. This section of the ms (lines 371-456) has undergone significant revision in two ways. First, we moved all results regarding FKH proteins further down in the ms (lines 458-744). In this way, we could now present, in a much more accessible format, predictions that if FDO1 deletion rescues... than FDO1 overexpression should be detrimental. These are indeed the findings and this strategy in flow persists throughout the remaining ms.

10. Line 256: "mcd1-1" should be eco1Δ rad61Δ?

Actually, mcd1-1 is correct as stated. Since we had to make this plasmid in-house (simple cloning), we still needed to test the resulting MCD1 over-expression plasmid in an mcd1-1 strain to validate it's efficacy. We revised the ms text (lines 243-246) to clarify this intermediate step. Note that we made additional MCD1 over-expression constructs that we also first tested in mcd1-1 ts strains to validate their efficacy.

We note here that the entire section regarding Mcd1 protein levels in eco1 rad61 cells is now the first section of the results (lines 168-258). Addressing this deficit in

Mcd1 levels in this first section facilitated a more linear flow in which each transcriptional network identified could then be queried for MCD1-dependent effects.

11. Line 267: Could you elaborate on START, MBF, and this first sentence to give a more detailed background?

Happy to do so. The text (lines 956-963) is revised to better explain this key transition and describe a bit more of the conundrum regarding a proposed role for MBF in cohesin pathways.

12. Line 292: Mbp1 instead of Mpb1.

Corrected - thanks (now line 1166)

13. Line 361: consistent "with" results from.

Corrected - thanks (now line 1476)

REVIEWER 2

The study proposes a pathway in which the mating type switching factor FDO1 regulates expression of the cohesion subunit MCD1, via the MBF complex. The *fdo1* gene was identified as a spontaneous revertant in a suppressor screen in the *eco1 rad61* background. A previous study by the same group described another hit from the screen, *cln2* (Buskirk et al 2022). The proposed pathway is novel and interesting, but the data, as presented in the current manuscript, do not support the conclusions. Additional experiments would be needed to be allowed to make the conclusions as stated.

We appreciate the reviewer's concluding that the findings are both 'novel and interesting' and have worked to address the concerns raised - described below.

Major points:

1. Fig 2: Mutation/deletion of *fdo1* rescues the temperature sensitive growth defects of *eco1 rad61* double mutants. However, deletion of *fkh1* or *fkh2*, encoding biochemical interactors of FDO1, does not rescue. Since these two genes are redundant in their cell cycle control role (according to ref 111, and as cited by authors), could they be functioning redundantly in this pathway, too? Does deletion of both *fkh1* and *fkh2* rescue the growth defects?

*We also considered the possibility that *fkh1* and 2 deletion effects were masked due to redundancy - a possibility that we did not pursue in the original ms due to the adverse effects obtained upon single deletion of FKH2. In response to the reviewer's*

*question, however, we generated new diploid strains in which we could test both for the ability to obtain *eco1 rad61 fkh1 fkh2* quadruple mutants (none of which were recovered) and *eco1 rad61 fkh1 fkh2* quadruple mutants in which a wildtype version of *ECO1* is integrated on a separate chromosome. The ability to obtain the *fkh1 fkh2* combination in *eco1 rad61* cells - but only if covered by an exogenous *ECO1* locus, suggests that the combination is lethal (as opposed to providing rescue). These results are included in lines 642-732 of the revised ms.*

*We extended our analysis to inquire whether the lethality produced by *fkh1 fkh2*, combined with *eco1 rad61*, might be based on reduced *Mcd1* levels. We tested this possibility and found that high-copy *MCD1* rescues *eco1 rad61 fkh1 fkh2* quadruple mutant cell inviability - although to only a very marginal extent in that this combination exhibits very severe growth defects. These findings are included as lines 733-746 of the revised ms.*

2. Fig 3: Overexpression of *FDO1*, *FKH1*, or *FKH2* all exacerbate the growth defects. This finding makes it very difficult to propose a way in which *FDO1* and one of the *FKH* may work together or against each other. The authors propose that these findings mean that *FKH1* and *FKH2* negatively impact cohesion pathways (lines 197-198). I disagree with this conclusion. The lethality could be due to some other function entirely unrelated to cohesin. An experiment to support the claim that *FKH1* and 2 regulate cohesion would be to check *MCD1* proteins levels (similar to the experiments in Fig 4).

*We agree that the genetic interactions are complex. That regulation of *MCD1* is indeed this complex is by itself an important point to raise to the scientific community. On the other hand, the reviewer raises a valid point that the exacerbated growth that results from *FKH2* overexpression might occur independent of *MCD1* expression. The experiment suggested by the reviewer is not likely to be informative given that we are already near the limit of *Mcd1* detection by Western blot even at permissive temperatures. Reducing *Mcd1* levels further (due to *FKH2* overexpression, for instance) is unlikely to produce a quantifiable effect.*

*The issue, however, is easily addressed in a different way (as suggested by Reviewer 3). If the adverse effect of *FKH2* or *FKH1* overexpression is truly independent of *MCD1* regulation (or confounded by additional genes), then co-overexpressing only *MCD1* should fail to rescue the deleterious effect produced by *FKH2* or *FKH1* overexpression. We tested this predication. Instead, our data reveals that co-overexpressing effectively rescues the deleterious effects caused by individual overexpression of *FKH1* or *FKH2* at 30C and also provides a limited rescue at 37C. These results are presented in lines 764-950.*

3. Fig 4: The authors argue that protein levels of the cohesion subunit *MCD1* decrease in *eco1 rad61* mutants, but that this defect is partially rescued by deletion of *fdo1*. The data presented does not support this conclusion, at least not the second part, that *MCD1* protein levels are higher in *eco1 rad61 fdo1* triple mutants compared to *eco1 rad61* double mutants. The difference is not statistically significant. The error bars, representing SEM, are huge, indicating the variability between replicates must be significant. Seeing the raw numbers (in a supplemental statistical table) might help

evaluate the data better. If the authors wish to make the conclusion that the triple mutant has higher protein levels than the double, the differences need to be statistically significant. Ideally, the experiment should be done using quantitative westerns. If not possible, at the very least the authors should do a dilution series for the proteins to demonstrate that they are in the linear range of detection. One of these methods may help reduce the variability between samples.

We agree that the original data regarding increased Mcd1 levels upon deletion of FDO1 had high SEM bars and realize that the change (while overt) was not statistically significant. In response to comments offered by both Reviewer 2 and 3 (see below), we separated our approaches regarding Mcd1 protein levels.

*In the first approach, we adopted the strategy offered by Reviewer 3 to test whether Mcd1 levels are reduced in *eco1 rad61* cells. To achieve this, we performed numerous new Western blots to better quantify the drop in Mcd1 levels in *eco1 rad61* cells, compared to wildtype cells. As suggested, we performed dilution series (100%, 50%, 25%, 12.5%) to ascertain a protein extract load obtained from wildtype cells (across all 3 biological replicates) that would fall into the linear range for both Mcd1 and Pgk1 (since Mcd1 levels are normalized to Pgk1). For wildtype extracts, the data indicates that Mcd1 and Pgk1 levels, across all 3 replicates, are roughly in linear range at 50% protein load. Below, we provide Western blot images and quantifications for both Mcd1 and PGK for all biological replicates (Figure 1).*

To test if Mcd1 levels are decreased in the eco1 rad61 null cells, we quantified Mcd1 protein levels in wildtype and eco1 rad61 null cells extracts - both diluted 50% as a concentration that gives reproducible changes in signal intensities. The results indicate that Mcd1 levels are decreased in the eco1 rad61 cells, compared to wildtype cell, and that this decrease is statistically significant (p value = 0.0031) (Figure 2 below). We note that the decrease in Mcd1 levels in the eco1 rad61 null cells, compared to wildtype cells, was statistically significant in the original manuscript as well.

We next tested the extent to which Mcd1 protein levels increase in eco1 rad61 cells that also harbor a fdo1 deletion. As before, we first established extract dilutions in which Mcd1 and Pkg1 levels both fall within a detectable range when assessed by Western blots (see dilution series in Figure 3 below).

We could now answer the second question - does FDO1 deletion, which rescues eco1 rad61 cell ts growth, result in increased Mcd1 protein levels? Using appropriately diluted extracts, our quantifications over 3 biological replicates reveals that deletion of FDO1 indeed increases Mcd1 protein levels in eco1 rad61 and that this change is statistically significant (p value = 0.0133). See Figure 4 below.

Note that these new experiments, and accompanying text, replace entirely that portion of the original ms (lines 359-370).

4. Fig 5: the data presented on this figure shows that MCD1 overexpression can in fact rescue the ts growth defects. If the argument is to be made that fdo1 deletion rescue due to an increase in MCD1 levels, the authors should compare MCD1 protein levels in eco1 rad61 fdo1 triple mutants to MCD1 overexpression in the double (on a quantitative western) and compare the relative protein levels to the effects on viability.

We anticipate that Mcd1 protein levels are quite high in the overexpressed strain, given the high copy number (40-60) achieved by a 2 μ m plasmid in haploid cells (Futcher 1986, J. Theor Biol; Chan et al., 2013, Plasmid). We submit that quantifying Mcd1 levels in this instance will not provide useful information regarding the level of Mcd1 that exceeds that required for cell viability. The issue is not how high Mcd1 protein levels go, but that cells retain a minimum threshold level of Mcd1 required to support cell viability. Both fdo1 deletion and MCD1 overexpression surpass that threshold. Prior studies from the Koshland lab suggest that the minimal level of Mcd1 required for viability is quite low - about 13% of wildtype Mcd1 (Heidinger-Pauli et al., 2010, Current Biology).

5. The Swi6 and Mbp1 results are again confusing. It again would help to see MCD1 protein levels in SWI6 and MBP1 overexpression lines to support the authors argument about the roles of these proteins in promoting MCD1 expression.

We are unsure of the source of confusion. We feel that the results are convincing that elevated MBP1 levels rescue eco1 rad61 ts growth (Figure 10B). We submit that the results also are clear that elevated levels of SWI6 abolishes the positive effect of

MBP1 when both are co-expressed (Figure 10C). Are the negative effects of SWI6 based on MCD1 regulation? As suggested by Reviewer 3, we performed additional experiments and now can show that MCD1 fully reverses the adverse effects produced by SWI6 overexpression (Figure 11). In combination, these findings provide strong evidence that Swi6 is a negative regulator of Mbp1-based transcription of MCD1. In support of this, we note prior findings that Swi6 (for instance, in association with Stb1) can inhibit MBF-dependent gene transcription (see for instance Costanzo et al., 2003 Mol Cell Bio). The model presented here - that Swi6 is a binding partner that can inhibit Mbp1-dependent expression of some genes - in this case MCD1, is thus not without precedence. The text has been revised to make clear this interpretation (lines 1237-1252).

On a more pragmatic note, we have nearly exhausted the polyclonal antibody supply of the Mcd1-directed antibody - a very generous gift from Dr. Vincent Guacci. Performing additional Western blots on cells overexpressing or deleted for the genetic combinations presented in this study would be a substantial burden requiring epitope tagging of Mcd1 in each and every genetic background. We submit that this will preclude publication indefinitely and ask that the reviewer not require this for publication

6. A final connection the authors try to make is that FDO1 represses mbp1 (and therefore its deletion leads to MBP1 overexpression and rescue). To support this argument, they would need to test whether deletion of both fdo1 and mbp1 (in an eco1 rad61 fdo1 mbp1 quadruple mutant) abrogates the rescue seen in eco1 rad61 fdo1 triple mutants.

In the revised ms, we chose not to pursue mechanisms through which Fdo1 might impact MBF activity. Trying to unambiguously connect each of these pathways with each other (either through competitive binding of promotor or generating various complex assembly in solution) is a biochemical project in its own right and beyond the current study parameters.

Minor points:

1. The writing needs work. There are several grammatical errors, and examples of improper choice of words. For example, the word "refractile" in line 259 is probably not the word the authors meant to use.

Thank you - we did take this opportunity to edit the manuscript, check for grammatical errors, and simplify the text as needed.

2. Background information could be presented better. For example, the abstract should state that MCD1 is a cohesion subunit. Line 98 should explain what NIPBL is.

We thank the reviewer for giving us the opportunity to revisit how the background information is presented. We shortened the Introduction by roughly 30%, which we feel produced an improved flow of ideas that we also limited to more tightly align to the topic at hand.

3. Fig 8: Panel A is not needed. The authors do not do any experiments on the *cln2* pathway. Panel B summarizes the model they propose. However, as explained above, additional experiments would be needed to actually propose this model.

We agree that a separate panel is not necessary. However, this study needs to be placed in the context of prior findings - especially those linked to MCD1 expression. In response, we collapsed components of panel A into panel B to generate one model figure (see Figure 13).

Reviewer #3 (Comments for the Authors (Required)):

Cohesin is a four-subunit complex (Smc1p, Smc3p, Mcd1p, Scc3p) that mediates multiple chromosome structure and function aspects. These include sister chromatid cohesion, condensation, DNA damage repair, transcriptional regulation, and chromosome segregation. Two cohesin regulators, Eco1p and Rad61p perform antagonistic functions. Eco1p acetylates Smc3p to stabilize cohesin binding to chromosomes, which enables the establishment of sister chromatid cohesion, whereas Rad61p destabilizes the chromosomal binding of non-acetylated cohesin. Given that only ~20% of cohesin is acetylated, a large pool of cohesin can be utilized for cohesin functions other than sister chromatid cohesion. All four cohesin sub-units are essential genes as is ECO1. In budding yeast, the deletion of RAD61 restores viability to *eco1Δ* mutants. However, the *eco1Δ rad61Δ* double mutant (*eco1Δ rad61Δ*) cells have numerous defects including highly defective cohesion, drug sensitivity and temperature sensitivity. Significant new insights on cohesin regulation have come from identifying and characterizing suppressors of these deficiencies of the *eco1Δ rad61Δ*. This paper describes a new set of suppressors that have the potential to provide important insights into cohesin regulation.

We appreciate that the reviewer found the study interesting and provided important insights.

The authors had previously conducted a genetic screen to identify suppressors of the *eco1Δ rad61Δ* temperature sensitivity and identified *cln2Δ* as a suppressor. Here, the authors identify and characterize an *fdo1* mutant as another such suppressor. Over-expression of FDO1 was toxic to *eco1Δ rad61Δ* cells, solidifying a genetic relationship. The authors examined Mcd1p levels and showed that it was reduced in *eco1Δ rad61Δ* cells, but this reduction was lessened in in the triple mutant (*fdo1Δ eco1Δ rad61Δ*), which provides evidence that suppression is due to increased Mcd1p levels. This conclusion is solidified by experiments showing that Mcd1p over-expression using a high copy plasmid bearing MCD1 provides an even more robust suppression than *fdo1Δ* does. They expand the genetic pathways by showing that over-expression of the transcriptional regulator MBP1 and its partner have opposite effects. MBP1 over-expression suppresses the *eco1Δ rad61Δ* temperature sensitivity, whereas SWI6 over-expression exacerbates the temperature sensitivity. This divergence suggests that

MBP1 and SWI6 have opposing actions with regard to controlling MCD1 expression. The authors present a model to explain suppression. They suggest that several transcription pathways control Mcd1p levels which can be altered to increase Mcd1p levels and promote suppression.

The experiments are generally well-controlled, and conclusions from most are consistent with the authors' interpretations. The data supports their model that MCD1 expression is under the control of different regulatory mechanisms and that perturbing this regulation to increase MCD1 expression suppresses the *eco1Δ rad61Δ* temperature sensitivity. However, the paper seems light in the amount of data presented, and the authors fail to present biological data to understand why increasing Mcd1p levels suppresses temperature sensitivity. As such, the paper is not yet suitable for publication in Genetics. A few simple experiments should be done to correct these deficiencies. First, the authors should test their model more directly by over-expressing MCD1 in *eco1Δ rad61Δ* cells deleted for either *fkh2Δ* or *fkh1Δ*.

*This experiment (and those broken out below) was a great way to test the validity of our prior interpretations. As outlined in the manuscript, only FKH2 deletion from *eco1 rad61* cells exacerbated their growth at 37C (with no overt impact at 30C). Importantly, overexpressing MCD1 fully restored the viability of *eco1 rad61* cells at 37C.*

The same goes for over-expressing MCD1 in *eco1Δ rad61Δ* cells also over-expressing SWI6 or one of the FKH1/2 or FDO1. The model would be strengthened if MCD1 over-expression does indeed suppress the temperature sensitivity or counteract the deleterious effects. It may be necessary to place MCD1 under control of an inducible promoter for such analyses to avoid any dominant inhibition of the native MCD1 promoter. It may be wise to add an intermediate temperature between 30{degree sign}C and 37{degree sign}C to detect more subtle effects.

*Separately, overexpression of either FKH1 or FKH2 produce exacerbated growth defects in *eco1 rad61* cells. Are these detrimental defects dependent on MCD1 expression? To address the reviewer's questions, we co-overexpressed MCD1 with both FKH1 and also FKH2. The results were surprisingly clear. MCD1 rescues not only the growth defects at 30°C, but rescue inviabilities at 37°C. As predicated by the reviewer, we did not expect a full rescue since the MCD1 exogenous construct still relied on the native MCD1 promoter. In this light, the amount of rescue is impressive (obviating the need for intermediate temperatures). We appreciate the opportunity to test our model and find that the FKH pathways are indeed critical for MCD1 regulation.*

A bigger deficiency is that the authors make no attempt to assess the biological basis for suppressing the *eco1Δ rad61Δ* temperature sensitivity. *eco1Δ rad61Δ* cells have defects such as severe cohesion defects, delays in the cell-cycle progression, as well as extreme sensitivity to drugs that damage DNA or destabilize microtubules. The paper would be greatly strengthened by comparing the phenotypes of the double mutant and the double mutant with suppressors (minimally *fdo1Δ* and MCD1 overexpression) by simple established tests for cohesion, drug sensitivity, and cell cycle progression. These

data would inform on which cohesin functions are rate limited by the levels of Mcd1 activity.

*What is clear from our current findings, which build on numerous prior studies, is that cells must sustain a level of Mcd1 protein for viability. Here, we reveal several novel transcriptional pathways required for cell viability and that appear to work through MCD1 expression (and thus protein levels). In terms of parsing out cohesin functions, we note prior work already described a stair-step set of defects that occur in response to reduced Mcd1 levels. Heidinger-Pauli et al (2010) assessed cells that contain 100%, 30% and 13% of Mcd1. They reported that cohesin's function in cohesion is functional even at 13% of wildtype levels and that this level supports cell viability. At 37°C, Mcd1 levels are not detectable in *eco1 rad61* cells, consistent with the notion that Mcd1 protein must fall below 13% and thus results in cell inviability. Condensation defects, however, arise at the 30% level of reduction and near normal levels of Mcd1 are required for efficient DNA repair. We submit that the time required to recreate these Koshland lab findings does not move the field forward in a significant fashion and submit that this is the current transcription-based study.*

INTRODUCTION:

The introduction reads more like a review than introduction to a Genetics paper dealing with suppression of the temperature sensitivity of *eco1Δ rad61Δ* cells. Too much detail and discussion of topics like TADs and looping when nothing in the results deals with TADs or looping. The entire 4th paragraph is about Roberts syndrome and end point they make is that cohesin functions not related to sister chromatid cohesion are important. The focus should be on the latter, especially data from *eco1Δ rad61Δ* double mutants, but Roberts syndrome can be briefly mentioned as one example of the importance of non-cohesin functions in human cells. There is basically no mention of phenotypes of *eco1Δ rad61Δ* mutants other than temperature sensitivity. The last important omission is that there is no mention that cohesin subunit levels fluctuate during the cell-cycle with expression increased in S phase. More importantly, Mcd1p is destroyed in anaphase and G1 phase so little remains in cells at these stages, making Mcd1p different from other cohesin subunits. This difference suggests that proper regulation of Mcd1p levels is of prime importance for cohesin function. Introducing the unique regulation of Mcd1p levels sets the stage for explaining their suppression of the temperature sensitivity of *eco1Δ rad61Δ* double mutants.

Two responses here. First, we agree that the Introduction was unwieldy and strayed off-topic. As noted in our response to Reviewer 2, the Introduction has been reduced by about a page of text. We feel that the flow and focus benefited greatly and thank both reviewers for the opportunity. Second, this reviewer, as well as comments from reviewer 1, prompted us to expand on the unique properties of Mcd1 and its potential role as a type of checkpoint. In response, the entire first paragraph of the Discussion (lines 1324-1368) is now focused on what we feel is both a new and exciting concept. Specifically in response to this reviewer, we included a description of the unique properties of Mcd1 in the Intro (lines 119-121) - which does help set up the basis for this study.

RESULTS

General for Figures:

A. It would be easier for the reader if the labels for 2 μ plasmids were placed on the same side as the genotypes rather than on the other side of the images.

Done - the episome was moved to reside next to the genotypes. Thanks.

B. The model presented discusses regulation of MCD1 gene expression. It would greatly strengthen the model if some RT-qPCR was used to quantify steady-state levels of MCD1 RNA levels, at least for conditions with the strongest effects.

*The transcription network documented in the current study is complex and one that we are pursuing - but under separate cover. For instance, two MBF binding sites reside upstream of MCD1 (both of which promote MCD1 expression) and ChIP places FKH protein also to a nearby site just upstream of MCD1. We expect to pursue RT-qPCR under conditions of reduced Mbp1 and Fkh1 and Fdo1 ... but we anticipate that integration of these pathways will make for a rather messy response. As outlined in the new Discussion, we also predict that Mcd1 protein degradation may be as much a culprit in rendering *eco1 rad61* cells inviable in response to thermic stress. Such effects won't be revealed by qRT-PCR, requiring that biochemical approaches (through which we can assess the role of various degradation pathways) will be needed in combination with qRT-PCR to generate a fuller understanding of Mcd1/MCD1 regulation. These are significant undertakings and beyond the scope of the current manuscript.*

Figure 4A. The FACS profiles are mis-labelled. The peaks should be labelled 1C and 2C for haploid genomes before and after DNA replication. A label of 1N and 2N means cells are a mix of haploids and diploids.

Correct - thanks. The revised manuscript contains the appropriate labels.

Figure 4B. The blots are problematic. There are two points to be made in these blots and they are best examined separately not in one blot. First, not all antibodies yield a linear signal. The authors should do a dilution series to avoid such issues. In the first blot they show Mcd1p levels are reduced in *eco1 Δ rad61 Δ* cells. Use WT at 100%, 75%, 50% and 25% and compare to 100% of *eco1 Δ rad61 Δ* cell extracts or some dilution of it. A second blot should be focused solely on a comparison of a dilution series of *fdo1 Δ eco1 Δ rad61 Δ* extracts to *eco1 Δ rad61 Δ* . After all, the key point is how does *fdo1 Δ* affect Mcd1p levels in *eco1 Δ rad61 Δ* cells. That is the reference, not how Mcd1p levels in *fdo1 Δ eco1 Δ rad61 Δ* compares to WT cells.

*Thank you for these suggestions - both the dilution series and separating out the impact of *eco1 rad61* on Mcd1 levels, versus how *fdo1* further impacts Mcd1 levels, was very helpful. In response (see detailed responses and figures above for Reviewer 2), we performed a dilution series of wildtype cell extracts at 100%, 50%, 25% and 12.5% for each of our biological replicates. We then re-performed each Western blot, using*

appropriately diluted extracts and found that the difference in Mcd1 levels between wildtype cell extracts and eco1 rad61 null cells extracts is statistically significant.

We then repeated this strategy to test whether the deletion of FDO1 increases Mcd1 protein levels, compared to eco1 rad61 cells that contain wildtype FDO1. Following results from this second dilution series across our biological replicates, we found that FDO1 deletion indeed increases Mcd1 levels and to a statistically significant level.

The extracts themselves seem a bit problematic. The authors claim band labelled by a star is a non-specific band. This seems unlikely as a non-specific band should be the same in all extracts but it is stronger in WT and the triple mutant. It appears to be an Mcd1p degradation product as are other bands with a slower mobility. The authors used a zymolyase treatment to generate protein extracts. That is dangerous as Mcd1p is easily degraded by several proteases. They should prepare new extracts using a TCA proposal to prevent degradation.

Early during our experiments, we performed side-by-side extraction methods (TCA versus zymolyase/SDS). In our hands, the zymolyase/SDS treatment (which does not involve the heat-build-up that we encounter during bead-beating in TCA, despite being performed in the coldroom with ice-bath breaks) was superior in reducing degradation. Note, for instance that some of the 'degradation' bands present in the TCA prep are absent in the zymolyase/SDS prep. This was the rationale for adopting the zymo/SDS strategy over the course of this study - a decision which we feel remains sound.

The identity of the major 'background' band (~60kD) remains unknown. One possibility, as noted by the reviewer, is that the non-specific band is cleaved Mcd1. In his 1997 paper, in which Dr. Guacci validated this Mcd1-directed antibody, he performed a set of arrests (alpha factor, HU and NZ) to ascertain Mcd1 levels across the cell cycle. He found that Mcd1 is largely absent during G1, peaks during S phase (HU) and persists at reduced levels into mitosis. We repeated these experiments and confirmed those results (not shown). Relevant here is that the background band was consistently present in all stages of the cell cycle - including during the G1 arrest when Mcd1 is not detectable (this is prior to MCD1 transcriptional upregulation). We submit that a reasonable explanation is that this band is indeed a background band, since the bulk of Mcd1 was degraded much earlier during anaphase onset and subsequently undergoes additional degradation steps. For the sake of argument, however, we cannot formally rule out that the band is a breakdown product.

DISCUSSION:

Line 312-313. It isn't precise to group FDO1, RAD61, CLN2 and ELG1 as negative regulators of cohesin. FDO1 and CLN2 negative regulators of cohesin transcription whereas RAD61 and ELG1 can be considered negative regulators of cohesin function.

We hope that the reviewer agrees that, at least in broad terms, the deletion of any one of those genes rescues cells diminished for Eco1 function - such that each is a 'negative regulator'. However, we agree with the reviewer that the mechanism of rescue is different - and that the original text failed to adequately make those differences clear. Unfortunate, since this is in part why this study is unique in identifying transcriptional pathways, and multiple pathways at that, through which MCD1 is regulated. In response, we modified the text (lines 1375-1441) to more accurately describe these different mechanisms.

Line 317-320. This is a bit general as well as inaccurate. Like *rad61* Δ , certain alleles of SCC3 and PDS5 can also suppress the inviability of *eco1* Δ , and these double mutants also have dramatic cohesion defects and drug sensitivities. ECO1 is not a cohesin subunit but Line 319 is written to suggest it is.

*We apologize for the glib inclusion of Eco1 in the sentence of 'other factors' that inferred cohesin subunits. We revised the sentence to more accurately reflect the roles of the players mentioned. We also revised the text (lines 1382-1487) to explicitly state that alleles of SCC3 and PDS5 can suppress *eco1* Δ inviability.*

Missing from the discussion: It has been shown that *rad61* Δ (i.e. *wpl1* Δ) has ~a 50% reduction in Mcd1p levels and is bound to chromosomes, yet cells are not temperature-sensitive. It is important to mention this distinction early: Reduced Mcd1p levels alone don't cause temperature sensitivity. Rather, the Ts-phenotype only occurs in the absence of Eco1p in a *rad61* Δ background.

*The reviewer brings up a point that we now realize was handled poorly in the original manuscript. We submit that the most important aspect of this study deals with the threshold of Mcd1 required for cell viability - and then identify how those levels are regulated. Toward this end, the Koshland lab previously reported that only about 13% of wildtype Mcd1 levels are required for yeast cell viability (Heidinger-Pauli 2010, Current Biology). At 30°C, *eco1* Δ *rad61* Δ cells retain ~40% Mcd1, compared to wildtype, which explains why they remain viable. We also tested *eco1* Δ *rad61* Δ placed at 37°C and found that Mcd1 levels fell precipitously (~4% of wildtype - see figure below). The band intensities for the mutant cells are difficult to quantify at this extremely low level. Given that simply elevating Mcd1 via exogenous high-copy plasmid is sufficient to rescue *eco1* *rad61* ts growth, we now had a powerful assay through which we can identify novel mechanisms through which MCD1 is controlled.*

As a separate issue, that rad61Δ is not lethal (nor ts), and that eco1Δ rad61Δ exhibit robust growth at 30°, were both previously reported and infers that both conditions provide for Mcd1 levels above the threshold required for cell viability. The extent to which Mcd1 protein levels are retained in viable cells, however, is not the focus of the current study. The reviewer, noting that rad61 cells exhibit a 50% drop is not lethal, also opens up a bit of a can of worms. For instance, other studies did not observe Wapl-dependent drop in Rad21 levels (see Gandhi 2006, Fig4 and Kueng 2006, Fig1F). In fact, chromatin-bound cohesin and increased rad21-gfp intensity was reported for cells deleted for wpl1 (Bernard 2008). Why these results differ in comparison to the Sutani 2009 study is not clear, beyond model systems. A similar discrepancy occurs as to whether Pds5 is required for stable cohesin binding (Panizza 2000) or is not required (Hartman 2000; Kulemzina 2012; Tong 2014; Choudhary 2022, etc). Pds5 is a really interesting player - but the above discrepancies are outside the focus of this current study. Instead, we focus on Mcd1 levels under which eco1 rad61 cells become inviable at 37C and then identify the pathways responsible for those changes in MCD1 expression.

Cheers

July 9, 2024

RE: GENETICS-2024-307170

Dear Dr. Skibbens:

I am pleased to accept your manuscript entitled "Fdo1, Fkh1, Fkh2 and the Swi6-Mbp1 MBF complex regulate Mcd1 levels to impact *eco1 rad61* cell growth in *Saccharomyces cerevisiae*" for publication in GENETICS, pending some very minor revisions. In particular, reviewer 3 provided some suggested edits for your manuscript that you should consider.

Please submit your revision along with a brief description of how you modified the manuscript in response to the reviewers' concerns and suggestions (which can be viewed at the bottom of this email). I expect you should be able to submit a revised manuscript within 30 days. A suitably revised manuscript will be acceptable for publication; I don't plan to send it out for re-review.

Thank you for submitting this story to Genetics. Enjoy the rest of the summer!

Sincerely,

Needhi Bhalla
Associate Editor
GENETICS

Approved by:
Jeff Sekelsky
Senior Editor
GENETICS

Reviewer comments:

Reviewer #1 (Comments for the Authors (Required)):

The authors have adequately answered my questions and comments.

Reviewer #2 (Comments for the Authors (Required)):

I believe the authors addressed all my concerns. I especially appreciate the added western blot experiments that provide strong support for the tested hypotheses.

Reviewer #3 (Comments for the Authors (Required)):

The authors have shorted and better focused the introduction, reorganized the results, modified the figure labeling and added some new experiments including Western Blots with better quantification. These changes have improved the manuscript so it is now suitable for publication in Genetics.

There are a few minor changes in the writing that the authors should consider.

Abstract:

Line 22: pluralize subunit. cohesin subunits

Introduction:

Lines 51-52: These lines are confusing. Change to: Cohesin tethers together sister chromatids from S phase through metaphase. These cohesins are stably bound until anaphase, when cohesion between sisters is lost to allow sisters chromatid to segregate.

Lines 72: Delete the line about Eco1/ESCO1 regulating loops in G1 as this study has no analysis of loops.

Lines 74: Add the word unacetylated.promotes dissociation of unacetylated cohesin cohesin from DNA.

Lines 86-87: what does early in the cell-cycle mean? G1, G1/S transition? Be more specific. End the sentence after you define this term. Then start a new sentence with "However, recent findings confirm..."

Results:

Throughout the manuscript when the authors refer to any protein, they don't write Mcd1p, Pgc1p but instead write Mcd1 or Pgc1. Please add the p at the end.

Lines 94-95: The title is awkward. "The temperature sensitivity of *eco1Δ rad61Δ* cells is due to reduced levels of Mcd1p" or "Reduced levels of Mcd1p in *eco1Δ rad61Δ* cells causes temperature sensitivity."

Lines 106-107: Change to: To test this possibility, we compared Mcd1p levels in wildtype and *eco1Δ rad61Δ* cells using quantitative Western Blots.

Lines 108: change during mitosis to during anaphase.

Lines 118: change Remarkably and add in S phase. "In contrast. Mcd1p levels were.....in S phase-arrested *eco1Δ rad61Δ* cells."

Lines 148: State what the mutation is here. "Only one gene, FDO1 was mutated in all segregants, a truncation of FDO1 (*fdo11-82Δ*). As the manuscript is written now this truncation just pops up later in dilution plating figure 3."

Lines 197-199: Remove the first phrase so the sentence starts with *eco1Δ rad61Δ* cells that harbor both MCD1 and FDO1..... (new Sentence) The fact that this rescue was less robust compared to MCD1 alone could reflect that FDO1 over-expression exerted some inhibition of the native MCD1 promoter on the high copy plasmid.

Line 233: Deleted the words "as expected".

Line 234: Add in the words "as expected" then split the sentence in half. As expected, *eco1Δ rad61Δ fkh2Δ* cells transformed with empty vector alone exhibited severe inviability at 37°C. In contrast, MCD1 overexpression partially rescued the inviability at 37°C. (don't use the word significant)

Lines 321-327: seem more like they are suited for the discussion. Can you simplify and shorten this for the results, then raise it in the same detail you did but in the discussion? I struggled to get the point you were making as a lot of new genes and mechanism are brought up here.

Figure legends

Legends for all figures with multiple diluting platings list the same comments after each part. For example, in Figure 10 "Temperatures and days of growth are indicated." is written 4x, once after each part of the figure. Just write at the end of each figure containing dilution plating, "Temperatures and days of growth are indicated for all panels". The same goes for the comment "Growth of tenfold serial dilutions" appears in each section. Write it once. I might change it to cells were plated in tenfold serial dilution on...

Discussion

Line 381: change "during G1 in yeast cells" to "in late G1...." or "at the G1/S transition.....").

Line 384. References for cells mutated SMC1, SMC3 and RAD61 reducing Mcd1p levels should be for *smc1 & smc3* (Boardman et al [2023] G&D 37:277-290) and for *rad61* mutants affecting Mcd1p levels (Bloom et al [2018] Genetics 208:111-124).

Line 387. Rather than saying it is only half of the checkpoint, why not say it is only active from S phase through metaphase. After all, the kinetochore/spindle checkpoint isn't active in G1 or before kinetochore duplication.

While a feedback affecting Mcd1p transcription is intriguing and possible,

I remind the authors that Mcd1p is destroyed when either Smc1p or Smc3p is destroyed, meaning that Mcd1p stability is dependent on it being in a trimer with Smc1p and Smc3p. This might preclude a buildup of Mcd1p in mutants even if MCD1 transcription was induced.

Line 394-397. This is a long sentence that is a bit confusing. Perhaps first state that Mcd1p over-expression suppresses viability of *eco1Δ rad61Δ* as well as (then list the specific mutants or double mutants from previous work you are referring to). Then the next sentence can bring up your checkpoint idea.

Dear Prof Bhalla,

We thank you again for the great news that this study is accepted for publication - pending minor revisions. Please again thank the reviewers for their efforts. Below, we provide detailed responses (embedded in *italics*) to reviewers' comments.

Reviewer #1 (Comments for the Authors (Required)):

The authors have adequately answered my questions and comments.

Thanks!

Reviewer #2 (Comments for the Authors (Required)):

I believe the authors addressed all my concerns. I especially appreciate the added western blot experiments that provide strong support for the tested hypotheses.

Thanks!

Reviewer #3 (Comments for the Authors (Required)):

The authors have shorted and better focused the introduction, reorganized the results, modified the figure labeling and added some new experiments including Western Blots with better quantification. These changes have improved the manuscript so it is now suitable for publication in Genetics.

Thanks!

There are a few minor changes in the writing that the authors should consider.

Abstract:

Line 22: pluralize subunit. cohesin subunits

Corrected

Introduction:

Lines 51-52: These lines are confusing. Change to: Cohesin tethers together sister chromatids from S phase through metaphase. These cohesins are stably bound until anaphase, when cohesion between sisters is lost to allow sister chromatid to segregate.

We follow grammatical rules of parallel construction. This paragraph compares two cohesin activities: those that occur during G1 and those that occur during S phase. Since the first segment starts with "During G1", the second should start with "During S phase". However, the remainder of the reviewer's suggestion seems appropriate and a revised version is now included in the manuscript.

Lines 72: Delete the line about Eco1/ESCO1 regulating loops in G1 as this study has no analysis of loops.

The reviewer's comment is a bit confusing in that they refer to 'this study'. The sentence in question is based on 3 separate references that together address cell cycle activity and extrusion. In reference 65 (Wutz 2020), the authors document changes in loop lengths in response to ESCO1 depletion (see Figure 7F). In reference 66 (Alomer 2017), the authors differentiate the cell cycle roles of ESCO2 from ESCO1, with ESCO1 playing the key role in acetylation before S phase, ie - during G1 (ESCO2 functions during S phase). In reference 67 (Van Ruiten 2022), the authors document enriched longer-range loop contacts in cells depleted of ESCO1 (and also confirm the G1 role of ESCO1). Respectfully, we decline to delete or revise the sentence in question.

Lines 74: Add the word unacetylated.promotes dissociation of unacetylated cohesin cohesin from DNA.

Agreed - thanks.

Lines 86-87: what does early in the cell-cycle mean? G1, G1/S transition? Be more specific. End the sentence after you define this term. Then start a new sentence with "However, recent findings confirm..."

Agreed. Revised to more clearly indicate the two separate concepts.

Results:

Throughout the manuscript when the authors refer to any protein, they don't write Mcd1p, Pgc1p but instead write Mcd1 or Pgc1. Please add the p at the end.

Our reading of the current literature, including studies by some of the leaders in the cohesin field, indicate that most researchers and journals no longer use (or at least require) the 'p' designation. This simplification is supported by eLife (Nasmyth 2023), Cell and also Molecular Cell (Srinivasan 2018; Chapard 2019), Nature Communications and also Nature Structural & Molecular Biology (Munoz 2022; Burmann 2019), PLoS Genetics (Orgil 2015), and even Genetics (Kuhl 2020). The use of 'p' occasionally appears, but this is based on the specific researcher. We prefer the easier to read form of dropping the p (which becomes highly repetitive in lists and unnecessarily clutters the text).

Lines 94-95: The title is awkward. "The temperature sensitivity of *eco1* Δ *rad61* Δ cells is due to reduced levels of Mcd1p" or Reduced levels of Mcd1p in *eco1* Δ *rad61* Δ cells causes temperature sensitivity.

It was a bit long. Thanks - shortened.

Lines 106-107: Change to: To test this possibility, we compared Mcd1p levels in

wildtype and *eco1Δ rad61Δ* cells using quantitative Western Blots.

Shortened - thanks.

Lines 108: change during mitosis to during anaphase.

Agreed - thanks for catching this. Now corrected.

Lines 118: change Remarkably and add in S phase. "In contrast. Mcd1p levels were.....in S phase-arrested *eco1Δ rad61Δ* cells.

Done

Lines 148: State what the mutation is here. "Only one gene, FDO1 was mutated in all segregants, a truncation of FDO1 (*fdo11-82Δ*)." As the manuscript is written now this truncation just pops up later in dilution plating figure 3.

Defining the mutation at this point helps a lot - thanks.

Lines 197-199: Remove the first phrase so the sentence starts with *eco1Δ rad61Δ* cells that harbor both MCD1 and FDO1..... (new Sentence) The fact that this rescue was less robust compared to MCD1 alone could reflect that FDO1 over-expression exerted some inhibition of the native MCD1 promoter on the high copy plasmid.

Thanks - we revised these lines to separate out the two concepts and more clearly state the likely nature of the attenuated rescue.

Line 233: Deleted the words "as expected".

Deleted

Line 234: Add in the words "as expected" then split the sentence in half. As expected, *eco1Δ rad61Δ fkh2Δ* cells transformed with empty vector alone exhibited severe inviability at 37°C. In contrast, MCD1 overexpression partially rescued the inviability at 37°C. (don't use the word significant)

Done (with minor additional revision).

Lines 321-327: seem more like they are suited for the discussion. Can you simplify and shorten this for the results, then raise it in the same detail you did but in the discussion? I struggled to get the point you were making as a lot of new genes and mechanism are brought up here.

Setting up the rationale for the experiment is critical - but we agree that this section can be simplified. The text has been shortened/revise to ensure the rationale is clear without all of the extraneous gene product details.

Figure legends

Legends for all figures with multiple diluting platings list the same comments after each part. For example, in Figure 10 "Temperatures and days of growth are indicated." is written 4x, once after each part of the figure. Just write at the end of each figure containing dilution plating, "Temperatures and days of growth are indicated for all panels". The same goes for the comment "Growth of tenfold serial dilutions" appears in each section. Write it once. I might change it to cells were plated in ten-fold serial dilution on...

That helps shorten the text - thanks.

Discussion

Line 381: change "during G1 in yeast cells" to "in late G1...." or "at the G1/S transition.....").

Done.

Line 384. References for cells mutated SMC1, SMC3 and RAD61 reducing Mcd1p levels should be for smc1 & smc3 (Boardman et al [2023] G&D 37:277-290) and for rad61 mutants affecting Mcd1p levels (Bloom et al [2018] Genetics 208:111-124).

The original references are valid, but we are happy to include these additional studies that confirm those prior results.

Line 387. Rather than saying it is only half of the checkpoint, why not say it is only active from S phase through metaphase. After all, the kinetochore/spindle checkpoint isn't active in G1 or before kinetochore duplication.

While a feedback affecting Mcd1p transcription is intriguing and possible, I remind the authors that Mcd1p is destroyed when either Smc1p or Smc3p is destroyed, meaning that Mcd1p stability is dependent on it being in a trimer with Smc1p and Smc3p. This might preclude a buildup of Mcd1p in mutants even if MCD1 transcription was induced.

As the reviewer is well aware, Mcd1 must be degraded and then expressed every cell cycle. If Mcd1 is indeed the focus of some surveillance mechanism, then both its degradation and transcription must be regulated. To make this notion clear to our readers, we respectfully decided to retain the framework used in the original version (that there are two mechanisms, or two halves, that together regulate Mcd1 levels).

We are currently pursuing new experiments to test the duality of this surveillance mechanism. The reviewer's model of complex stability (degradation) is one possibility - but given that our entire study reveals a complex and overlapping array of transcriptional programs, it is important to test the extent to which both are at play.

Line 394-397. This is a long sentence that is a bit confusing. Perhaps first state that

Mcd1p over-expression suppresses in viability of $eco1\Delta rad61\Delta$ as well as (then list the specific mutants or double mutants form previous work you are referring to. Then the next sentence can bring up your checkpoint idea.

We provided a reference for the other specific mutants and simplified the text.

Thanks!

Bob Skibbens and Gurvir Singh

July 19, 2024

RE: GENETICS-2024-307170R1

Prof. Robert V. Skibbens
Lehigh University
Biological Sciences
111 Research Dr
Bethlehem, Pennsylvania 18015

Dear Dr. Skibbens:

Congratulations! We are delighted to inform you that your manuscript entitled "Fdo1, Fkh1, Fkh2 and the Swi6-Mbp1 MBF complex regulate Mcd1 levels to impact *eco1 rad61* cell growth in *Saccharomyces cerevisiae*" is acceptable for publication in GENETICS. Many thanks for submitting your research to the journal.

To Proceed to Production:

1. Format your article according to GENETICS style, as discussed at <https://academic.oup.com/genetics/pages/general-instructions>, and upload your final files at <https://genetics.msubmit.net>.
2. Your manuscript will be published as-is (unedited-as submitted, reviewed, and accepted) at the GENETICS website as an Advanced Access article and deposited into PubMed shortly after receipt of source files and the completed license to publish. Please notify sourcefiles@thegsajournals.org if you do not wish to publish your article via Advanced Access.
3. We invite you to submit an original color figure related to your paper for consideration as cover art. Please email your submission to the editorial office or upload it with your final files. You can submit a small-sized image for evaluation, and if selected, the final image must be a TIFF file 2513px wide by 3263px high (8.375 by 10.875 inches; resolution of 600ppi). Please avoid graphs and small type.

If you have any questions or encounter any problems while uploading your accepted manuscript files, please email the editorial office at sourcefiles@thegsajournals.org.

Sincerely,

Needhi Bhalla
Associate Editor
GENETICS

Approved by:
Jeff Sekelsky
Senior Editor
GENETICS

note: Please add jnls.author.support@oup.com and genetics.oup@kwgglobal.com (or the domains @oup.com and @kwgglobal.com) to your email program's "safe senders" list. You will be contacted by both at various points during the production process.